# Comparative G-Protein-Coupled Estrogen Receptor (GPER) Systems in Diabetic and Cancer Conditions: A Review

**DOI:** 10.3390/molecules27248943

**Published:** 2022-12-15

**Authors:** Aliyu Muhammad, Gilead Ebiegberi Forcados, Abdurrahman Pharmacy Yusuf, Murtala Bello Abubakar, Idris Zubairu Sadiq, Isra Elhussin, Md. Abu Talha Siddique, Suleiman Aminu, Rabiatu Bako Suleiman, Yakubu Saddeeq Abubakar, Babangida Sanusi Katsayal, Clayton C. Yates, Sunila Mahavadi

**Affiliations:** 1Center for Cancer Research, Department of Biology, Tuskegee University, Tuskegee, AL 36088, USA; 2Department of Biochemistry, Faculty of Life Sciences, Ahmadu Bello University, Zaria P.M.B. 1044, Nigeria; 3Biochemistry Division, National Veterinary Research Institute, Vom P.M.B. 01, Nigeria; 4Department of Biochemistry, School of Life Sciences, Federal University of Technology, Minna P.M.B. 65, Nigeria; 5Department of Physiology, Faculty of Basic Medical Sciences, College of Health Sciences, Usmanu Danfodiyo University, Sokoto P.M.B. 2254, Nigeria; 6Centre for Advanced Medical Research & Training (CAMRET), Usmanu Danfodiyo University, Sokoto P.M.B. 2254, Nigeria

**Keywords:** diabetes mellitus, malignancy, G-protein-coupled estrogen receptor, biosignaling, therapeutics

## Abstract

For many patients, diabetes Mellitus and Malignancy are frequently encountered comorbidities. Diabetes affects approximately 10.5% of the global population, while malignancy accounts for 29.4 million cases each year. These troubling statistics indicate that current treatment approaches for these diseases are insufficient. Alternative therapeutic strategies that consider unique signaling pathways in diabetic and malignancy patients could provide improved therapeutic outcomes. The G-protein-coupled estrogen receptor (GPER) is receiving attention for its role in disease pathogenesis and treatment outcomes. This review aims to critically examine GPER’ s comparative role in diabetes mellitus and malignancy, identify research gaps that need to be filled, and highlight GPER’s potential as a therapeutic target for diabetes and malignancy management. There is a scarcity of data on GPER expression patterns in diabetic models; however, for diabetes mellitus, altered expression of transport and signaling proteins has been linked to GPER signaling. In contrast, GPER expression in various malignancy types appears to be complex and debatable at the moment. Current data show inconclusive patterns of GPER expression in various malignancies, with some indicating upregulation and others demonstrating downregulation. Further research should be conducted to investigate GPER expression patterns and their relationship with signaling pathways in diabetes mellitus and various malignancies. We conclude that GPER has therapeutic potential for chronic diseases such as diabetes mellitus and malignancy.

## 1. Introduction

Diabetes Mellitus (DM) is estimated to have affected 540 million people worldwide in 2021, accounting for approximately 10.5% of the world’s population [1]. Unfortunately, predictions show that this number will increase in the coming years, with USD 966 billion to be spent on diabetes-related healthcare expenditure and about 1 million mortality cases to be recorded annually [1]. DM is a broad term that encompasses type 1, type 2, and gestational diabetes, in addition to other secondary types such as Cushing syndrome, acromegaly, etc., all of which are conditions characterized by impaired glucose homeostasis. The most prevalent is type 2 diabetes mellitus (T2DM), which is more prevalent in the elderly population [2]. DM and its complications are associated with significant economic implications for patients and caregivers [1]. Various side effects have been reported among diabetic patients who use glucose-lowering drugs, necessitating research into alternative therapeutic targets [3].

Malignancy is another disease of global interest for which it is projected that about two million new malignancy cases and over six hundred thousand malignancy-related deaths will occur in 2022 in the United States alone [4]. The most diagnosed malignancies are prostate, breast, lung, and colorectal cancer. However, in many parts of the world, gastric cancer (GC) is a leading cause of cancer-related morbidity and is the fourth most prominent cause of malignancy-associated mortality [5]. Despite the various therapeutic interventions available for the management of GC, the disease has a high incidence and poor prognosis in central and south America, eastern Europe, and east Asia largely due to chemoresistance, disease recurrence, and metastasis [5]. The prominent observation that GC treatment outcomes differ across racial and ethnic groups has led to the recommendation that a mono approach to gastric cancer management is unsuitable, requiring precision-medicine strategies [6]. Epigenetic modifications in other forms of malignancy have also been suggested to contribute to the varied disease pathogenesis observed for GC patients [7]. Just in the United States alone, new cases of breast (287,850), prostate (268,490) and lung (236,740) cancers have been predicted in 2022 [4]. The burden from these estimated new malignancy cases will require efficacious treatment options, especially if there is a better understanding of their pathogenesis as it relates to biomolecule-induced signaling pathways, like that of the G-protein-coupled estrogen receptor (GPER) systems.

Current research is examining the role of G-protein-coupled estrogen receptor (GPER) in disease pathogenesis, as well as interventions that could target this protein for disease management [8]. In vivo studies have shown that estrogen binding to its receptors mediates a variety of functions other than reproduction. For instance, GPER knockout mice reproduce but exhibit metabolic defects [8]. Similarly, GPER deficiency is associated with an increased risk of insulin resistance, obesity, dysregulated homeostasis, and altered glucose and lipid metabolism. GPER expression was lower in gastric cancer tissues compared to normal tissues [9]. Indeed, it was demonstrated that GC patients with low levels of GPER mRNA had lower overall survival, suggesting that downregulation of GPER expression was associated with poor prognosis. On the contrary, for ovarian cancer, GPER overexpression correlates with poor treatment outcomes. GPER overexpression is also associated with tamoxifen resistance and lower survival rates for ER-positive breast cancer patients [10]. In contrast, for triple-negative breast cancer, elevated GPER expression is useful in inhibiting angiogenesis and metastasis in triple-negative breast cancer [11]. Although GPER dysregulation is evident in most malignancies with inconsistencies in the manner with which it is expressed, there is a dearth of information on the mechanisms underlying these alterations, necessitating further research. Thus, this review critically examines the literature on the comparative role of GPER in DM and malignancies, highlighting research gaps that need to be addressed and pointing to the potential of GPER as a therapeutic target for the management of both DM and malignancy.

### Methodology

This review is a scoping type that utilized articles from PubMed, Scopus, and google scholar databases. The key words used for the search are Diabetes mellitus, malignancy, G-protein-coupled estrogen receptor, biosignaling, epigenetics and therapeutics. The search engine covered original research articles published from 2004 to 2022.

## 2. GPER System in DM

Hyperglycemia, the hallmark of DM, results from impaired insulin production, secretion, or signaling and is a prerequisite for the vascular complications of DM [12,13,14]. Decades of research have documented the roles of the classical estrogen receptors (ERα and ERβ) in the metabolism and homeostasis of glucose and lipids but the importance of GPER in these processes is coming to light [15]. Recent evidence suggests that many aspects of glucose-mediated insulin synthesis and release from β-cells as well as insulin signaling in peripheral tissues involve rapid non-genomic GPER-mediated estrogen signaling [15,16]. In addition, GPER expression in the pancreatic islets is associated with the enhancement of overall survival and prevention of apoptosis of β-cells [17,18]. Similarly, the estrogen-mediated protection against oxidative and endoplasmic reticulum stress-induced β-cell apoptosis following glucotoxicity involves GPER as well as genomic signaling [19,20,21]. Regarding insulin resistance and type 2 DM (T2DM), GPER is differentially expressed in the livers of T2DM patients and its signaling could be associated with hepatic insulin resistance [22].

Existing evidence suggests a lower risk of weight gain, obesity, metabolic syndrome, and diabetes in mice and human premenopausal females compared to males and postmenopausal females; this protection has been attributed to estrogens, especially E2 (17β-estradiol) [23,24]. Moreover, the antidiabetic roles of E2 and synthetic GPER agonists such as G-1 were significantly abolished in GPER knockout human cell lines and murine models, and GPER deficiency correlated with a higher incidence of insulin resistance, adiposity, and dysregulated homeostasis and metabolism of glucose and lipids [15,23]. Furthermore, a recent study reported that exclusive GPER activation by G1 is involved in the homeostasis of glucose and lipids in E2-deficient females and diet-induced obese (DIO) male models of mice, but, unlike E2, the compound had no feminizing or bone-mineral preserving effects [25]. These findings, therefore, imply that, unlike the classical estrogen receptors, GPER signaling could have a predominant role in energy metabolism compared to its reproductive and other physiological roles.

It is noteworthy that elevated levels of estrogens and exposure to exogenous GPER antagonists could contribute to the pathogenesis of DM [26,27,28]. Thus, GPER modulation in physiological and diabetic conditions is complicated and depends on several factors, including sex, age, the extent of GPER expression, and the type as well as concentrations of ligands (agonist or antagonist) that interact with the receptor [17,23,27,29]. The role of GPER in diabetic conditions has been excellently reviewed [15,26,30]. Here we give an overview of the recent advances in the role of GPER signaling in the pathogenesis of DM and attempt to establish new molecular mechanisms and therapeutic targets/options.

### 2.1. GPER Deficiency and Signaling in the Pathogenesis of DM and Its Complications

Accumulating evidence in murine models has proven that downregulated or abolished GPER expression results in impaired homeostasis and metabolism of glucose and lipids, thereby increasing the risk of cardiometabolic diseases such as DM and malignancy [30,31]. For instance, a study reported that female GPER knockout mice were obese and glucose-intolerant at 6 months of age whereas their male age-mates were only obese and insulin-resistant but with normal glucose levels [32]. The researchers, however, reported the manifestation of other metabolic syndrome phenotypes including dyslipidemia, glucose intolerance, and inflammation in the GPER knockout male mice at a later age (1–2 years) [32], indicating that GPER deficiency affects both sexes but in an age-dependent manner.

It is noteworthy that, although E2 at physiological concentrations elevate insulin sensitivity, abnormal elevation of E2 could contribute to the pathogenesis of T2DM. For example, it has been previously reported that high levels of E2 coupled with elevated insulin levels in insulin-resistant individuals could synergistically exacerbate the condition by upregulating aromatase P450 activity and GPER-mediated dysregulation of the PI3K/Akt signaling [26]. Moreover, elevated E2 during pregnancy (especially at the late stage) is associated with insulin resistance and mechanistically involves GPER-mediated proteolytic cleavage (ectodomain shedding) of the insulin receptor [33]. The evidence thus provides insight into the possible involvement of GPER signaling in the pathogenesis of gestational diabetes mellitus (GDM). Apart from elevated levels of estrogens, exposure to environmental (nanomolar) concentrations of endocrine-disrupting chemicals such as bisphenol A (BPA) could worsen the risk of developing DM [28]. A recent study suggests that BPA exposure induces apoptosis of human and rat insulin-producing cells by disrupting the protective effects of ERα, ERβ, and GPER signaling [27]. The researchers reported a kind of crosstalk involving the three ERs where BPA-mediated GPER activation reduced ERαβ dimerization leading to the subsequent loss of the anti-apoptotic activities of both receptors on β-cells [27]. This result signifies that the protective role of estrogens against β-cell apoptosis is mediated by crosstalk involving all three ERs.

Importantly, GPER signaling could also worsen the risk of diabetic complications for diabetic patients. For example, the endothelial/vascular protective role of aldosterone, E2, and G1 is mediated by GPER signaling under normal physiological conditions. This protection is, however, significantly reduced in a mouse model of obesity and T2DM, although, in this model, GPER gene and protein expressions were upregulated in this model [34]. Based on the findings of this study, GPER-mediated aldosterone signaling increased the response to contractile signals in the mesenteric-resistant arteries of the mice due to diabetes-induced endothelial damage whereas the vasodilator role of aldosterone was decreased/lost [34]. Because diabetic conditions result in elevated serum aldosterone and estrogen levels [26,34], we suggest that the loss of the endothelial protective roles of these hormones in diabetic models could be attributed to dysregulated GPER signaling following their elevation. In contrast, another study revealed that, under high glucose conditions, only ER-α activation, but not GPER or ER-β, inhibits human aortic vascular smooth muscle cell proliferation [35]. These observations imply that the involvement of estrogen signaling in the pathogenesis of the vascular complications of DM involves the activation of a pool of signaling pathways involving genomic and non-genomic responses.

### 2.2. GPER Signaling in Diabetic Complications: The Protective Roles of Estrogens and Their Mimetics

The discovery of GPER as an estrogen receptor involved in rapid non-genomic cellular signaling paved the way for the elucidation of its interactions with other non-genomic signaling pathways. Indeed, there is an overlap or crosstalk among GPER signaling and some key downstream pathways of the insulin/IGF system such as the PI3K–Akt, MAPK/ERK, and the mTOR signaling pathways. These pathways are involved in the regulation of cell growth, apoptosis, and metabolism of nutrients such as carbohydrates and lipids, and are thus implicated in the pathogenesis of DM and related sequelae [36,37]. Accordingly, the antidiabetic actions of estrogens and related GPER agonists could be attributed to the GPER-mediated modulation of these pathways. For instance, in a type 1 diabetic (T1D) model of mice with pancreatic islet transplant (PIT), E2, E2α (17α estradiol), and G1 significantly improved islet engraftment by increasing β-cell mass, inhibiting hypoxia-induced β-cell apoptosis as well as stimulating angiogenesis and β-cell vascularization in the islet transplanted mice [38]. According to the researchers, estrogens exert this effect by activating GPER signaling; the effect was retained even after the treatment had been stopped [38]. This observation suggests that estrogens minimize islet graft rejection in T1D patients with PIT by modulating GPER signaling, and the overall effect could be enhancement of islet angiogenesis and maintenance of proper blood and oxygen delivery such that hypoxia-induced β-cell apoptosis could be minimized.

A mechanistic study reported that GPER activation by E2 resulted in increased insulin secretion in pancreatic β-cells by increasing intracellular calcium mobilization under low- and high-glucose conditions [24]. According to the authors, the observed increase in insulin secretion resulted from GPER-mediated crosstalk between the stimulatory EGFR-ERK and the inhibitory PI3K/Akt signaling pathways, with the EGFR-ERK pathway having a predominant role in the process [24]. Another study reported that E2 increased GLUT-2-mediated insulin secretion in pancreatic islets and stimulated glucose uptake and catabolism in peripheral tissues by increasing GLUT-2 protein expression via upregulating GPER and its downstream targets in the Akt/mTOR pathway [39]. In a similar study, genistein, the most abundant phytoestrogen in soybeans, promoted glucose uptake in broiler chickens and human cell lines by upregulating GLUT-2 protein levels as well as increasing the activities and gene expression levels of enzymes involved in glucose oxidation via activation of a GPER-mediated cAMP/PKA-AMPK signaling pathway [40]. In addition, a GPER-mediated E2/β-catenin/T-cell factor (E2/β-cat/TCF) signaling cascade is shown to upregulate the hepatic expression and downstream action of the hepatic fibroblast growth factor 21 (FGF21) in a transgenic model of mice with a dominant-negative mutation of TCF7L2 [41]. FGF21 is a hormone that improves glucose and lipid homeostasis in various animal models; male mice, but not females, exhibited increased hepatic and plasma triglyceride (TG) levels, indicating a failure to improve lipid homeostasis by the hormone [41]. The finding implies that the hypolipidemic effect of the hormone could be improved by GPER-mediated estrogen signaling, which mechanistically involves GPER-mediated activation of the β-cat/TCF transcription complex. Moreover, because β-catenin is a downstream effector of the Wnt signaling pathway [41], Wnt activators could also be involve in regulating the hepatic expression of the hormone.

GPER signaling may be involved in the therapy of DM and in the protection against or reversal of vascular complications associated with the disease. In ovariectomized db/db mice, GPER, but not ERα activation, restored high glucose-induced vascular endothelial dysfunction by improving endothelial nitric oxide synthase (eNOS)-catalyzed nitric oxide (NO) synthesis [42]. Similar results were obtained for high glucose-treated human aortic cell lines and the underlying mechanism involves GPER-mediated inhibition of the RhoA/Rho kinase-induced suppression of eNOS activity [42]. RhoA activation (due to hyperglycemia or high glucose treatment) suppresses eNOS activity by reducing the phosphorylation of the eNOS stimulatory site (Ser 117) and enhancing the phosphorylation of its inhibitory site (Ser 495); the effect is reversed by G1 and E2-mediated GPER stimulation [42]. The authors also reported that similar signaling is involved in the estrogen-mediated alleviation of oxidative stress and apoptosis in treated endothelial cell lines [42]. The study thus provides a mechanistic basis for the GPER-mediated role of estrogen signaling in NO-induced vasodilation and endothelial protection, which is reportedly compromised in diabetic conditions. In another study, icariin, a phytoestrogen, attenuated high glucose-induced glomerular fibrosis in human and rat mesangial cells by inhibiting the transforming growth factor β (TGF-β)-mediated accumulation of the extracellular matrix (ECM) proteins, collagen type IV, and fibronectin, via GPER activation [43]. The authors suggested that icariin-induced GPER signaling led to decreased TGF-β-mediated activation of Smad2/3 and ERK 1/2 by phosphorylation, leading to inhibition of their target genes and subsequently reduced expression and accumulation of the aforementioned ECM proteins [43]. In a similar study, icariin attenuated oxidative stress-induced mitochondrial dysfunction and inhibited the apoptosis of mice podocytes (caused by high glucose treatment) via GPER-mediated upregulation of the anti-apoptotic protein (Bcl2) [44]. Moreover, a recent mechanistic study suggests that the GPER-mediated protective role of icariin against the high glucose-induced accumulation of EM proteins and oxidative stress in diabetic nephropathy could involve Nrf-2 signaling, because icariin augments the expression, nuclear translocation, and downstream signaling of Nrf2 in high glucose-treated human mesangial cell lines [45]. Figure 1 summarizes the GPER downstream signaling pathways involved in the homeostasis of glucose and lipids under high glucose conditions as well as the mitigation of high-glucose-induced vascular complications of DM following ligand activation.

### 2.3. Molecular Mechanism of GPER Signaling in DM: The Potential Role of Epigenetic Modifications

Although there is a dearth of literature on the role of epigenetic modifications in the modulation of GPER signaling in diabetic conditions, emerging evidence suggests the involvement of microRNAs in the epigenetic regulation of GPER-mediated estrogen signaling in DM and its vascular complications [46]. For instance, E2 at its physiological concentration induced the upregulation of miR-30b-5p expression in human umbilical vein endothelial cells by activating GPER [47]. In contrast, human urinary exosomal miRNA profiling revealed downregulated miR-30b-5p expression in T2DM patients with diabetic nephropathy. Putting these observations together, we suggest that diabetes-induced perturbations in the GPER-mediated estrogen signaling are implicated in the pathogenesis of diabetic nephropathy; and the proposed molecular mechanism could involve miR-30b-5p downregulation. This warrants further investigation.

Evidence also suggests that early-life exposure to BPA increases the risk of DM in adulthood by reprogramming the fetal epigenome [48,49,50,51,52]. However, despite the wealth of evidence depicting BPA as a GPER modulator in DM, the involvement of GPER in BPA-mediated epigenetic reprogramming is not well established for diabetic conditions. Nevertheless, a mechanistic study found that GPER stimulation due to a short-term BPA exposure impaired glucose-induced insulin secretion by pancreatic β-cells by downregulating miR-338 and its target gene, *Pdx-1*, a master gene that regulates β-cell survival and glucose-stimulated insulin secretion [53]. Additionally, considering the role of GPER in the regulation of survival, prevention of apoptosis, and mediation of glucose-induced insulin secretion in β-cells under hyperglycemic and diabetic conditions, we postulate that BPA exposure abolishes this beneficial role by stimulating GPER to facilitate the epigenetic reprogramming of genes downstream of the receptor.

As mentioned earlier, there is currently little or no evidence for the epigenetic dysregulation of GPER expression for diabetic conditions in the presence or absence of ligands. However, in diabetic models, expressions of the classical estrogen receptors have been epigenetically altered. For instance, ERα expression is epigenetically dysregulated in the placental tissues of pregnant women with GDM in a cell type and gender-specific manner [54]. Based on the study, ERα expression was upregulated in a particular placental cell type (extravillous trophoblast) obtained from the tissues of GDM patients compared to non-diabetic controls, and the result positively correlated with hypomethylation of the receptor’s gene promoter [54]. In contrast, in the endothelial cells of the decidual tissues of non-diabetic controls, ERα upregulation was only observed only in pregnancies with male fetuses and not females, and this gender-specific effect was abolished in the GDM group [54], suggesting that GDM is involved in this variation. In another study, diabetes-induced poor wound healing in rats was reported to be associated with the epigenetic downregulation of the ERβ and its target genes; the effect was ameliorated by pterostilbene-mediated reversal of the epigenetic changes [55]. The researchers reported elevated repressive histone methylation marks (H3K9me2 and H3K27me3) around the ERβ gene’s promoter and the promoters of its targets (some of the genes involved in accelerated wound healing) in hematopoietic stem cells and peripheral blood mononuclear cells of the affected rats compared to non-diabetic controls; following pterostilbene treatment, these epigenetic changes were completely restored in all of the cell types following pterostilbene treatment [55]. We, therefore, suggest that based on these observations, the classical ERs are epigenetically dysregulated in diabetic conditions and that reversal of these diabetes-induced epigenetic changes could be a promising therapeutic strategy. Recently, a study reported the upregulation of GPER and ERβ protein levels in broiler chickens and human cell lines following genistein supplementation [40]. Accordingly, genistein increases β-cell proliferation, prevents β-cell apoptosis, enhances glucose-stimulated insulin release, and has the potential to induce epigenetic changes in obese and diabetic models [56,57]. Thus, the molecular mechanism of genistein-mediated upregulation of GPER and ERβ expressions as well as the expression of some genes/proteins downstream of GPER could involve epigenetic rewiring. Moreover, considering the lack of evidence showing the direct epigenetic modulation of GPER expression in diabetic conditions, further research on this aspect could unveil molecular mechanisms and therapeutic potentials.

## 3. GPER System in Various Malignancies

The estimated new malignancy cases will necessitate effective treatment options, mainly if a better understanding of their pathogenesis related to biomolecule-induced signaling pathways, such as the GPER systems, is gained. GPER expression patterns in various cancers are highly complex and now debatable, with some cancers showing upregulated GPER expression patterns and others showing downregulated or even inconclusive GPER expression patterns. GPER, for example, is overexpressed in seminomas, melanomas, some ovarian cancers, lung cancers (NSCLC), insulin-resistant endometrial cancer models, and the vast majority of breast cancer models (particularly triple negative breast cancer, TNBC). We provided sequential explanation, critical appraisal, knowledge gaps and way forward about GPER systems versus malignancies in the subsequent sections.

### 3.1. GPER in Testicular Germ Cell Cancer

Testicular germ cell cancer (TGCC), the most common solid cancer among young men, expresses both classical (ERβ) and non-classical (GPER/GPR30) estrogen receptors [58]. GPER is often overexpressed in seminomas but not in non-seminomas [58,59]; this overexpression tends to stimulate seminoma cell proliferation by activating ERK1/2 and protein kinase A. An in vitro analysis using JKT-1 cell lines (derived from a human testicular seminoma) revealed that the binding of endocrine disruptors (EDCs) such as bisphenol A (at a low concentration) to GPER induces seminoma cell proliferation [60,61]. Estradiol-17β conjugated to bovine serum albumin has been reported to stimulate JKT-1 cell proliferation via activation of the PKA pathway with rapid phosphorylation of cAMP response element binding protein (CREB) through GPER [60]. Further, in TCam-2 seminoma cell line, estradiol, acting through GPER-cAMP/PKA/CREB signaling, induces ERα36 isoform expression, in TCam-2 seminoma cell line, leading to increased cell proliferation [62]. GPER overexpression in human testicular seminomas is associated with ERβ downregulation, causing a switch in estrogen responsiveness [63]. In the TCam-2 cell line, 17β-estradiol induces the activation of ERK1/2 and increases c-Fos expression through the GPER-associated ERβ downregulation. According to a cohort study, there is a genotype-phenotype association between GPER SNPs and seminomas [59]. Two of these SNPs (SNP rs3808350 located in the 5′-regulatory region and SNP rs3808351 in the 5′-untranslated region) located in the promoter region of the GPER gene allow them to modify the gene’s expression pattern [59], as evident in overexpression of GPER previously reported for TGCC [58,60,64]. Thus, a better understanding of the role of GPER in tumorigenesis of TGCC would open perspectives to identify new therapeutic targets.

### 3.2. GPER in Breast Cancer

Breast cancer is a complex disease for which a plethora of factors contribute to alterations that favor the initiation and progression of the disease [65]. Although estrogen-associated signaling has been associated with canonical ER-α and ER-β signaling, current research is focusing on the role of GPER downstream signaling in breast cancer pathogenesis and response to therapy [66]. GPER is expressed in breast cancer tissues with an abundance of about 50–60%. Activated GPER activates signaling proteins, including the EGFR, phosphatidylinositol 3 kinase (PI3K), mitogen-activated protein kinases and focal adhesion kinase along with other proteins linked to the regulation of cell growth and division [67].

Due to variations in the pathophysiology and treatment of the different breast cancer subtypes, studies have broadly examined GPER expression in ER-positive and ER-negative subtypes [68]. For patients, GPER expression correlated with estrogen and progesterone receptor status and poor response to therapy [69]. For ER-positive breast cancer patients, GPER expression is associated with tamoxifen resistance and lower survival rates [10]. GPER activation and downstream signaling increases the expression and membrane localization of ATP- binding cassette subfamily G member 2 (ABCG2), which is implicated in multi-drug resistance; inhibition or knockdown of GPER attenuated these effects [66]. However, GPER activation by G-1 reduced cell proliferation, induced cell cycle arrest at M-phase, and enhanced apoptosis in MCF-7 ER-positive breast cancer cells [70]. The authors suggested that GPER activity could be affected by epigenetic factors, as GPER expression was inactivated by promoter methylation.

In triple-negative breast cancer (TNBC), GPER expression is increased and GPER-mediated signaling contributes to TNBC invasiveness, metastasis, and recurrence through mechanisms involving the activation of focal adhesion kinase (FAK) and signal transducer and activator of transcription 3 (STAT3), which mediates alterations in expression of target genes [67]. Experiments with the MDA-MB-231 TNBC cell line showed that in the presence of bisphenol A, a GPER ligand, there was increased GPER expression and activation of FAK and extracellular signal-regulated kinase 2 (ERK2), which mediate invasiveness and metastasis in TNBC [71]. The role of GPER in TNBC proliferation was deduced from a study that examined the interaction of GPER and Na^+^/H^+^ exchanger regulatory factor 1 (NHERF1) on the proliferation of MDA-MB-231 cells [72]. The interaction of NHERF1 with GPER inhibited GPER-mediated activation of ERK1/2 and suppressed the proliferation of TNBC cells [72]. In TNBC cells treated for 96 h with 200 nMgefitinib, an inhibitor of EGFR that is transactivated by GPER, there were downregulation of GPER and suppressed cell proliferation [73].

In contrast, GPER expression is protective in TNBC, as MDA-MB-231 xenograft tumors treated with the GPER agonist G1 showed inhibited angiogenesis and metastasis via mechanisms involving the inhibition of NF-kB and IL-6 signaling [11]. Additionally, MDA-MB-231 and MDA-MB-468 TNBC cells treated with the G-1 agonist for GPER activation showed inhibited cell growth via mechanisms involving induction of cell cycle arrest at G2/M and increased caspase-3 mediated apoptosis [70]. For MDA-MB-231 TNBC cells, GPER activation reduced their proliferation and invasiveness of the cells [74]. Clinical studies with 135 Chinese TNBC patients found that GPER expression, determined by immunohistochemistry, positively correlated with favorable treatment outcomes and negatively correlated with lymph node metastasis [75]. A similar study accomplished in the United Kingdom involving 1,245 primary invasive breast cancer patients found that high GPER expression was associated with smaller tumors and lower tumor grades [76]. However, the study did not report on the ER status of the patients. The different patterns of GPER expression observed in different breast cancer phenotypes are presented in Table 1.

#### 3.2.1. GPER Localization and Predisposition to Breast Cancer Subtype

Although the expression and localization of GPER has long been debated, several studies have revealed that, in breast cancer cells, GPER is detectable both on the surface of the cell membrane and intracellularly within the endoplasmic reticulum and Golgi apparatus [77,78]. A study revealed a predominant cytoplasmic GPER expression in 189 primary invasive breast carcinomas (19.3%); a predominant nuclear GPER expression was observed in 529 cases (53.9%) [79]. Using murine knockout models, GPER overexpression and localization in the plasma membrane have been shown to be essential events for breast cancer progression [80]; its absence in the plasma membrane has been reported to have an excellent long-term prognosis for ERα+ breast cancer patients treated with tamoxifen [78]. The plasma membrane localization of GPER is associated with an increased risk of death for metachronous contralateral breast cancer [81]. Additionally, plasma membrane-localized GPER correlates with poor prognostic markers such as high Ki67 and a triple-negative subtype. Additionally, GPER found within the endoplasmic reticulum likely contributes to normal estrogen physiology as well as pathophysiology due to its specific binding to estrogen, resulting in intracellular calcium mobilization in the nucleus [82]. Furthermore, cytoplasmic GPER expression in breast carcinomas is associated with non-ductal histological subtypes, better histological differentiation, luminal A and B subtypes, low tumor stages, and a better clinical outcome [79]. In contrast, the expression of nuclear GPER in breast carcinomas is associated with triple-negative subtypes with less favorable clinical outcomes [79]. In estrogen-stimulated, breast cancer-associated fibroblasts (CAFs), there is a peculiar GPER translocation to the nucleus where it targets genes such as c-FOS and CTGF leading to increased expression of these genes [83]. A P16L polymorphism present in CAFs affects the N-glycosylation status of GPER, which promotes its nuclear localization [84]. This alternate subcellular localization of GPER in CAFs and potentially in the carcinoma cells themselves may modify the action of these cells and affect tumor progression [84]. Tamoxifen-resistant MCF-7 breast cancer cells maintain a proliferative response to estrogen by promoting the translocation of GPER to the cell membrane as well as it signaling [68]. Moreover, long-term tamoxifen treatment has been reported to facilitates the translocation of GPER to cell membranes, resulting in abnormal activation of the EGFR/ERK signaling pathway, which enhances communications between tumor cells and their microenvironment [85]. This reveals that GPER subcellular localization influences its function in the progression and prognosis of breast carcinomas.

#### 3.2.2. GPER Target Genes in the Progression of Breast Cancer Malignancies

Estrogenic effects have been ascribed to the nuclear estrogen receptors (ERα and ERβ), which function as transcription factors binding to the regulatory response elements in the promoters of target genes [86]. However, estrogen also triggers non-genomic, rapid cellular events through a seven-transmembrane GPER referred to as GPR30 [87]. GPER is expressed both in the plasma membrane and in the endoplasmic reticulum [81]. It is involved in the proliferation, migration, chemoresistance, and metastasis of breast cancer [88]. The binding of estrogen and related compounds to GPER activates multiple intracellular signaling pathways, including the MAPK, adenylyl cyclase, and PI3K signaling pathways through transactivation of the (EGFR) [89]. Therefore, GPER is required for EGFR and ERK activation by epidermal growth factor [90]. It is noteworthy that GPER nongenomic signaling events can result in long-term transcriptional changes and a broad range of responses among a variety of cell types [59].

In a study by Filardo et al., 17β-estradiol (E2)-triggered rapid activation of ERK1/2 in breast cancer cells that correlated with GPER expression [91]. It was demonstrated that GPER-dependent ERK activation occurred via transactivation of the EGFR through matrix metalloproteinase activity and integrin α5β1, which triggered the extracellular release of heparan-bound epidermal growth factor (HB-EGF) [92]. Cellular signaling resulting in GPER activation leads to ERK1/2 phosphorylation [93]. A study by Pupo et al. confirmed that ERK1/2 phosphorylation by bisphenol A was abolished by silencing GPER expression, suggesting that GPER is required for ERK1/2 activation [94]. ERK1/2 has multiple substrates such as transcriptional regulators and steroid hormone receptors, which mediate several biological processes, such as migration, proliferation, angiogenesis, and invasion [95]. In mammary tumors, ERK1/2 substrates are mostly hyperactivated because of the high mutation rate of approximately 30% in ERK1/2 members. Pupo’s findings further revealed that bisphenol A transactivates the promoter sequence of *c-FOS*, early growth response 1 (*EGR-1*), and connective tissue growth factor (CTGF) and hence stimulated mRNA expression of these genes and increased their protein levels [94].

GPER mediates the transactivation of the EGFR to the MAPK signaling axis in response to E2 [91]., GPER is associated with the modulation of calcium (Ca^2+^), cAMP, and phosphatidylinositol 3-kinase (PI3K) [96]. GPER-triggered rapid signaling events have been determined to regulate gene expression of BCL2 and cyclin D1 [97], and *EGR-1* [98,99]. Madeo and Maggiolini [98] revealed that GPER is exclusively expressed as an estrogen receptor in mammary cancer-associated fibroblasts and induces the expression of C-FOS, cyclin D1, and CTGF in response to E2, as confirmed at both the mRNA and the protein levels, resulting in the promotion of proliferation. In response to estrogen, GPER is recruited to chromatin at target genes such as c-FOS, PI3K, Src, FAK, BCL2, cyclin D1, connective tissue growth factor (CTGF), and early growth response 1 (EGR1) [98,100]. Overexpression of Focal adhesion kinase (FAK) and Src kinases has been observed in breast cancer cell lines and tumors. In MDA-MB-231 cells, bisphenol A activates FAK, Src, and ERK2 via GPER in MDA-MB-231 cells [101]. Bisphenol A also stimulates AP-1 and NF-κB-DNA binding via an Src and ERK2-dependent pathway. Src family kinases regulate cell cycle progression, growth, survival, and migration. Src has been associated with EGFR transactivation via phosphorylation at specific tyrosine residues. FAK and Src regulate cell motility, an important component of focal adhesion and cell migration. Breast tumors and cell lines have increased activity of src kinases and FAK, which are associated with tumor growth and metastasis [100]. E2 and GPER selective agonist G1 activates FAK, ERK2, and Akt1 via GPER in TNBC MDA-MB-231 and TNBC SUM-159 cells [67]. E2 and tamoxifen activate FAK and cause migration in endometrial cancer cells via GPER [102]. FAK is a 125 kDa protein tyrosine kinase that is found in focal adhesion, and is involved in a variety of biological processes such as spreading, differentiation, proliferation, apoptosis, migration, invasion, survival, and angiogenesis [103].

In breast cancer-associated fibroblasts, the GPER agonist G1 induced CYP19A1 gene expression and increased E2 production via the GPER/EGFR/ERK pathway which in turn promoted proliferation and cell-cycle progression [104]. In mammary CAFs, GPER/EGFR/ERK signaling has been claimed to upregulate the expression of EGR1, CTGF, C-FOS, and cyclin D1, resulting in proliferation enhancement in mammary CAFs [94,99]. Bisphenol A to stimulates the proliferation and migration of SKBR3 cells and carcinoma-associated fibroblast (CAFs) through GPER [94]. Growth factors including EGF, CTGF, transforming growth factor a/b, and insulin-like growth factor to regulate GPER expression and are regulated by GPER activation [98,105]. GPER expression correlates with upregulation of aromatase gene (an enzyme involved in estrogen synthesis) upregulation [94,106] (Table 2).

A well-recognized GPER agonist, genistein, increases breast cancer-associated aromatase expression and activity in vitro [106]. Moreover, in MCF-7R cells, GPER/EGFR/ERK signaling upregulates β1-integrin expression and activates downstream kinases, which contributes to cancer-associated fibroblast-induced cell migration and the epithelial-mesenchymal transition [107] (Table 2). Integrins are focal adhesion receptors that mediate adhesion between the actin cytoskeleton and the extracellular matrix, with various components, including scaffolding proteins, GTPases, kinases, and phosphatases [114]. Integrins mediate signal transduction between the tumor cell and its microenvironment and β1-integrin and coordinates some cellular processes such as inflammation, proliferation, adhesion, and invasion [115]. Hypoxia upregulates CTGF expression in a HIF-1α-dependent manner; however, GPER expression is required for the HIF-1α-mediated induction of CTGF by hypoxia [108]. CTGF primarily modulates and coordinates signaling responses involving components of the extracellular matrix [97].

Activation of GPER by its specific agonist significantly inhibits interleukin-6 (Ile-6) and vascular endothelial growth factor A (VEGF-A) by suppressing NF-κB binding to the Ile-6 gene promoter. Proangiogenic factors including Ile6 and Ile8 are highly expressed and necessary in the growth of triple-negative cancer cells [116]. These proteins activate quiescent microvascular endothelial cells and induce them to form tube structures [117]. The vascular endothelial growth factor (VEGF) is a chemoattractant and proliferative cytokine that initiates angiogenesis in tumor development [75]. E2 and the GPER-selective ligand G-1 triggered a GPER/EGFR/ERK/c-fos signaling pathway that leads to increased VEGF via upregulation of HIF1α [110]. In a mouse xenograft model of breast cancer that ligand activated GPER can enhanced tumor growth and the expression of HIF1α, VEGF, and the endothelial marker CD34 [110]. In addition, a mechanistic study revealed that E2-mediated GPER activation upregulated HOTAIR (lncRNAs highly expressed in primary breast tumors) gene expression in the TNBC cell lines (MDA-MB-231 and BT549) as well as in peripheral blood mononuclear cells and cancer tissues from breast cancer patients through the suppression of miR-148a [111]. Therefore, HOTAIR could be viewed as a potential therapeutic target in breast cancer; and its level in primary tumors is a powerful predictor of metastases and death [111]. GPER stimulation activates yes-associated protein 1 (YAP1) and transcriptional coactivator with a PDZ-binding domain (TAZ), two homologous transcription coactivators and key effectors of the Hippo tumor suppressor pathway, via the G**α**q-11, PLC**β**/PKC, and Rho/ROCK signaling pathways. TAZ is required for GPER-induced gene transcription, breast cancer cell proliferation, migration, and tumor growth [118]. Chrysin-NPs inhibit the proliferation of MDA-MB-231 cells via induction of GPER expression, and this action suppresses matrix metalloproteinases (MMPs) and NF-kB expression [119]. In breast cancer, the expression levels of MMPs are higher than in normal breast tissues. MMPs are key enzymes responsible for ECM breakdown and are involved in the regulation of cell growth, migration, angiogenesis, and invasion [120]. A study using breast SkBr3, colorectal LoVo, hepatocarcinoma HepG2 cancer cells, and breast cancer-associated fibroblasts suggested that GPER is involved in regulating fatty acid synthase (FASN) expression and activity in cancer cells and cancer-associated fibroblasts, which contributes to cancer progression [112]. FASN is needed in cancer cells for synthesis of fatty acids, which in turn are used for energy production, cellular membranes, signaling molecules, and membrane protein anchors [121]. In SkBr3 cancer cells and CAFs, miR-338-3p suppresses gene expression and proliferative effects induced by E2 through GPER. miR-338-3p is involved in cancer development through inhibition of the expression of certain genes at both transcriptional and post-transcriptional levels [122]. For SkBr3 breast cancer and HepG2 hepatocarcinoma cells, E2 and the selective GPER ligand G-1 induce miR144 expression and downregulate the onco-suppressor Runx1 through GPER [113].

### 3.3. GPER in Lung Cancer

Lung cancer is the most common cause of cancer death worldwide and the second most frequent cancer and the major cause of cancer death in the United States [123]. Gene expression studies reveal that the expression of GPER is considerably greater in human non-small cell lung cancer cell lines (NSCLC) as compared to normal lung cells [124]. This receptor appears to promote NSCLC cell proliferation by modulating the circ-NOTCH1/YAP1/QKI/ /m6A methylated NOTCH1 pathway, and in this way, the GPER upregulates circNOTCH1, blocks YAP1 phosphorylation, and increases YAP1-TEAD transcriptional regulation on QKI [125] (Figure 2). The notch signaling pathway could thus a GPER targets in lung cancer. In lung cancer, heavy metals can activate the GPER and related signaling pathways. For example, cadmium chloride and sodium arsenite have been shown to enhance the proliferation of human lung adenocarcinoma cell lines by activating GPER/ERs/ERK/MAPK signaling pathways [126]. Bisphenol A (BPA), a ubiquitous pollutant with endocrine disrupting effects, may increase carcinogenesis susceptibility as treatment with BPA promotes the migration and invasion of human lung cancer cells, an effect linked to morphological changes and matrix MMP-2 and MMP-9 overexpression mediated by GPER [127] (Figure 2). Furthermore, numerous estrogenic-synthetic chemicals activate GPER, including herbicides, plasticizers and pesticides [128,129]. According to a genotoxic study, BPA caused enhanced DNA damage in the Hep-2 cell line and oxidative damage in the MRC-5 cell line [130]. Co-exposure to BPA and doxorubicin (DOX) in both cell lines MRC-5 and Hep-2 demonstrated that BPA acts as a DOX antagonist and interacts with MRC-5 cells, a GPER-expressing cell line, which appears to be cell-type dependent, resulting in a non-monotonic response curve [130].

When estrogen binds to both ER and GPER, a signal is generated, which leads to increased cell proliferation [131] (Figure 2). The activity and expression levels of enzymes involved in estrogen production are upregulated in some cases. For instance, there was an increase in the mRNA level of estrogen metabolic enzymes (17β-HSD type 1 and aromatase) in the lungs of COPD patients compared to controls, implying that estrogen exposure has a role in the etiology of COPD [132] and possibly lung cancer [133]. Since the GPER has been associated with development of NSCLC, a selective GPER inhibitor, for example, the G15 has been linked to inhibition of this type of cancer development by reversing E2-induced cell proliferation [134]. Thus, GPER dynamics could be a factor in lung carcinogenesis.

### 3.4. GPER in Gastric Cancer

Gastric cancer is the sixth most frequent cancer worldwide and the third leading cause of cancer-related death [135]. This type of cancer develops from malignant cells in the stomach lining, and is divided into two topographic subtypes: cardia gastric cancers, which develop closest to the esophagus, and non-cardia cancers, which develop farther away [136]. When compared to normal tissues and cells, the GPER mRNA and protein levels were lower in gastric cancer tissue and in cells examined, demonstrating that, in gastric cancer, decreased GPER expression predicts for poor prognosis [9]. GPER expression stimulates G-1-induced anticancer effects by increasing cleaved poly ADP-ribose polymerase, caspases activity, and ER-stress signaling-induced expression of PERK, ATF-4, GRP-78, and CHOP [137]. Accordingly, the stresses generated in the ER by G-1 promote gastric cancer cell death. Knockdown of GPER-1 inhibited GC cell proliferation, migration, and invasion, as well as downregulated mesenchymal markers (vimentin and N-cadherin), upregulated the epithelial marker E-cadherin, and suppressed expression of the transcription factors [138]. The GPER polymorphisms of rs3808350 and rs3808351 were strongly connected with cancer propensity, notably for the Asian population, and, in malignant tissues, cancer severity and advancement were correlated with rs3808351 and GPER expression in malignant tissues [139]. By implication, these results show that GPER is a player in genetic polymorphisms associated with gastric cancer.

### 3.5. GPER in Colon/Colorectal Cancer

Colorectal cancer is a term used to describe two types of cancer that affect the large intestine: colon and rectal cancer [140]. Colon and colorectal cancers are similar in that they both originate in the large intestine. However, although colon cancers originate in the colon, colorectal cancers originate in the rectum [140]. The implication of this is that since the rectum has proximity to other organs, metastasis is more common with rectal cancer than colon cancer [141]. Men are more prone to rectal cancer than women [141]_._ Additionally, due to factors requiring further research, the rectal mucosa is far more prone to carcinogenesis and malignancies than the colon mucosa [140]. The risk for developing colorectal cancers is higher in Europe and America when compared to Africa and Asia, with a projected increase of 60 to 70% in colorectal cancer- associated deaths by 2035 [141].

The challenges associated with the management of colorectal cancers have necessitated research into the pathophysiology of these diseases [142]. Estrogen has function in gastrointestinal motility including colon motility [143]. Estrogen receptors are expressed all through the gastrointestinal tract, pointing to estrogen-mediated signaling in gastrointestinal and colon function [144]. Thus, experiments have examined the role of GPER in colorectal cancers [145,146]. A study involving adult female mice showed GPER expression in colonic myenteric neurons, and reduced colonic transit and increased nitric oxide production on the administration of a GPER blocker (G15) and increased nitric oxide production on administration of GPER agonist (G1), thereby pointing to GPER mediated effects on colonic motility [147]. Immunohistochemistry results from in vivo studies using mice and human colons, showed GPER expression with inhibition of muscle contractility observed when G1 and estradiol were administered in vivo [145]. In colon tissues of colorectal cancer patients, GPER expression was observed to be significantly downregulated compared to matched normal tissues [148]. Further lower GPER expression in the tumors of patients was associated with poor survival rates when compared to those expressing higher GPER levels in tumor tissues [148].

Considering these observations, further studies are needed to examine the potential of GPER agonists as a therapeutic target. Such studies should take into consideration the age of patients, ancestry, and stage of cancer. The results from such studies would provide insight into the potential of GPER as a therapeutic target for colorectal cancers

### 3.6. GPER Signaling in Renal, Liver, and Pancreatic Cancer

Recently, immunohistochemical staining techniques employing an anti-human GPER monoclonal antibody (20H15L21) revealed, among 31 tumor entities, GPER protein expression in human renal, hepatocellular, and pancreatic tumors [149]. However, the authors reported that only pancreatic and renal cell carcinomas have showed moderate to strong and potentially clinically relevant GPER expression patterns [149], which projects that receptor as a potential diagnostic option or therapeutic target in these cancers. Moreover, emerging in silico data are beginning to uncover the molecular mechanism of GPER-ligand interactions. A phenylalanine cluster at the GPER active site is essential for GPER recognition by its agonists and antagonists and this molecular scaffold is proposed to be the binding site of some synthetic tetrahydroquinoline derivatives that inhibit the proliferation of human renal, liver, and pancreatic cancer cell lines in vitro [150]. The finding thus provides insight into the potential role of GPER in the pharmacological treatment of these cancers.

#### 3.6.1. Renal Cancer

Although it accounts for just about 2% of the global cancer burden, renal cell carcinoma (RCC) is the deadliest and the third most common urinary malignancy [151,152]. RCC cell lines exhibit high GPER protein expression levels, and GPER activation in these cells is involved in RCC metastasis [152,153]. For instance, by activating GPER, aldosterone promotes RCC metastasis in murine RCC cell lines in a dose-dependent manner [153]. In a recent study, G1-mediated pharmacological activation of GPER promoted the migration and invasion of RCC cells, which mechanistically involves the GPER-mediated upregulation of matrix metalloproteinase-9 (MMP-9) via PI3K/Akt signaling [152]. We could thus infer from these observations that GPER promotes RCC metastasis by modulating the expression and function of extracellular matrix proteins involved in the EMT of RCC cells via the PI3K/Akt signaling axis and perhaps by yet-to-be-discovered signaling pathways.

#### 3.6.2. Liver Cancer

Regarding liver cancer, a study reported that GPER could serve as a prognostic biomarker in human hepatocellular carcinoma (HCC) patients because its protein expression is downregulated in the cancerous tissues of patients compared to normal controls, and this downregulation is associated with poor overall survival [154]. The authors also reported that G1 inhibited tumor growth of HCC xenografts by activating GPER/EGFR/ERK signaling [154], indicating that the pharmacological activation of GPER could be a viable therapeutic strategy for HCC patients. The finding implies that GPER downregulation occurs during HCC development and hence, we speculate that epigenetic mechanisms could be involved. In another study, E2 and G1-mediated GPER stimulation inhibited leptin-induced proliferation of HCC cell lines via GPER/ERK/STAT3 signaling [155]. The outcome of this experiment suggests that GPER is involved in mitigating the oncogenic properties of elevated leptin and thus, reduce the susceptibility of obese individuals to HCC. Moreover, there are downregulated GPER mRNA and protein levels in tumor specimens obtained from HCC patients undergoing curative liver resection [156]. According to the authors, GPER knockout accelerated tumorigenesis in mice by enhancing inflammatory responses and fibrosis, but no significant changes were observed in HCC cell lines with GPER knockdown although the growth of these cells was inhibited by G-1 despite being GPER deficient. The authors thus concluded that the tumor-suppressive role of GPER in HCC patients could be an indirect effect involving the mitigation of inflammation rather than a direct impact on the HCC cells [156]. Furthermore, tamoxifen could prevent hepatic fibrosis in HCC patients by promoting mechanical deactivation of ECM protein synthesis as well as suppressing actomyosin-dependent contractility and mechanosensing of external tissue stiffness via a GPER-mediated RhoA/myosin signaling in hepatic stellate cells (HSCs) [157]. Tamoxifen-mediated GPER activation downregulated RhoA levels, which in turn reduced the phosphorylation of MLC-2 (an actomyosin regulatory protein that controls ECM contractility in HSCs) and also suppressed YAP activation; actinomycin and YAP activation are necessary for the development of fibrosis in HCC tumors [157]. The authors also reaffirmed that tamoxifen-mediated modulation of GPER/RhoA/myosin signaling impeded the adaptability of HCC cells to hypoxia-induced necrosis by downregulating *HIF-1α* gene expression [157]. The studies thus provide insight into the possible involvement of GPER in the mechanical remodeling of the tumor microenvironment and prevention of hepatic fibrosis for HCC patients. Accordingly, in a previous mechanistic study that dehydroepiandrosterone (DHEA)-mediated GPER activation promoted tumorigenesis in human HCC cell lines by upregulating miR-21 transcription via a GPER/Src/EGFR/MAPK signaling cascade [158]. Based on the research outcomes, the signaling cascade increases miR-21 transcription by promoting the recruitment of the androgen receptor, c-Fos, and c-Jun to its promoter [158]. It is noteworthy that miR-21 is oncogenic and overexpressed in HCC cells, and, according to the authors, the RNA in human HCC tumors is actively involved in the downregulation of tumor suppressor genes such as *PDCD4* [158]. This study, therefore, suggests that epigenetic mechanisms are involved in the GPER-mediated DHEA-induced HCC tumorigenesis.

#### 3.6.3. Pancreatic Cancer

Evidence suggests that GPER proteins are expressed in tissue samples of pancreatic ductal adenocarcinoma (PDAC) patients, and the survival probability in these patients increases significantly with an increase in GPER expression [159,160]. Furthermore, G-1-mediated GPER activation inhibited the EMT and, therefore, the metastasis of PDAC cell lines by suppressing the contractility and mechanotransduction capacity of the cells via a GPER/RhoA/myosin-2-mediated deactivation of YAP [159]. Similarly, tamoxifen-induced GPER signaling increases vascularization and inhibits fibrosis and the hypoxic response in PDAC cell lines via the GPER/RhoA/myosin/HIF-1A axis [161,162]. These observations support the role of GPER in the RhoA-mediated regulation of the mechanical properties of the tumor microenvironment of HCC cells as reported [157] and thus present RhoA as a promising target in the GPER-mediated pharmacological treatment of these cancers. Further, G-1-mediated GPER activation inhibited the proliferation of human and murine PDAC cell lines by enhancing their susceptibility to immune destruction via downregulating the protein expression of c-Myc and its target, PD-L1, which are required for the proliferation, invasion, and escape of PDAC cells from immune surveillance [160]. We therefore infer from this observation that GPER signaling could enhance immunotherapy for PDAC patients.

### 3.7. GPER in Endometrial Cancer

Endometrial cancer is a uterine cancer that starts in the lining of the uterus, and represents the most common gynecological cancer in the developed world [163]. This type of cancer starts in the layer of cells that make up the uterine lining (endometrium). According to research, GPER interacts with the autocrine motility factor (AMF) and promotes PI3K signaling, promoting endometrial cancer (EC) cell growth [164]. GPER also regulates diacylglycerol kinase (DGK) activity, which is necessary for 17-estradiol (E2)-induced proliferation, motility, and anchorage-independent development of the Hec-1A endometrial cancer cell line [165]. Expression of endometrial estrogen receptors (ERa, ERb, and GPER) is essential for regular menstrual cycles and eventual pregnancy [166], and therefore altered expression of these receptors may lead to endometriosis, endometrial hyperplasia, and endometrial carcinoma which may affect many women of reproductive age [166]. Endometrial cancer has about 6-fold lower GPER mRNA expression than normal endometrium, and G-1 treatment slowed the growth of the GPER-positive cell lines but had no effect on a GPER-negative cell line [167]. Thus, the expression of GPER in the cell could help anticancer agents to slowing down cancer cell growth and exhibit greater efficacy. The expression of Egr-1 in breast and endometrial cancer cells is stimulated by E2 and hydroxytamoxifen (OHT) and mediated via GPER/EGFR/ERK signaling and in model cell lines (breast and endometrial cancer cells). Egr-1 is necessary for the proliferative effects generated by E2, OHT, and G-1 [99]. GPER expression was higher in the endometrial tissues of EEC participants with insulin resistance, and insulin, through epigenetic regulation, increased TET1 and GPER expression [168]. Low GPER mRNA causes loss of GPER protein from the primary tumor to metastatic lesions, which ultimately translates to disease progression [169]. In endometrial cancer, miR195 inhibits cell migration and invasion by targeting GPER and inhibiting the epithelial-mesenchymal transition; both GPER expression and the AKT/PI3K signaling pathway were found to be involved [170].

A novel tamoxifen analogue referred to as STX activates GPER, resulting in stimulation of GPER and possibly elevating estrogen levels, as well as activating the PI3K and MAPK pathways, causing endometrial cancers proliferation [171]. GPER mediates estrogen-stimulated activation of ERK and PI3K via matrix metalloproteinase activation and subsequent EGFR transactivation (Figure 2), and thus ER-targeted medicinal drugs act as GPER agonists in this regard, and pharmacological inhibition of GPER activity may inhibit estrogen-mediated endometrial tumor growth [77]. Tamoxifen has been associated with endometrial cancer proliferative effects and lymph node metastases in users, and this proliferative impact is driven by the GPER/EFGR/ERK/cyclin D1 pathway, which might be blocked by GPER silencing [172]. The impact of insulin-induced TET-1 expression and subsequent expression of GPER has been studied. Accordingly, insulin increased TET1 expression, and TET1 was involved in upregulating GPER expression and activating the PI3K/AKT signaling pathway, thus allowing insulin to stimulate EC cell proliferation through TET1-mediated GPER upregulation [173]. Despite having a diverse expression of estrogen biosynthesis and metabolic genes, various EC models have been discovered to produce estrogens only via the sulfatase route. These cells, however, expressed all of the major genes involved in the production of hydroxy estrogens and estrogen quinones, as well as their conjugation [174]. The involvement of TET1 in the GPER system, specifically in EC, points to the fact that GPER epigenomics could be among the underlying molecular mechanisms of EC proliferation.

### 3.8. GPER in Melanoma Skin Cancer

Melanoma, the most dangerous skin cancer which occurs due to the abnormal growth of melanocytes that are responsible for skin color [175]. As per the American Cancer Society’s estimation for melanoma in the United States, about 99,780 new melanomas will be diagnosed, and about 7650 people are expected to die of melanoma in 2022 [176]. In addition, racial disparities also exist for melanoma as it is 20 times more common in whites than in African Americans. There are several risk factors linked to melanoma, including sun (UV) exposure, family history, gender (being male), age, skin color, and genetic mutations (predominantly CDKN2A) [177,178]. In search of new treatment methods, it was observed that specific gender (female), multiple records of pregnancies, and early age of first pregnancy are correlated with a reduction in melanoma [179,180]. Therefore, it was thought that sex hormone signaling must be involved. Between 1996 and 1998, four distinct laboratories discovered GPR30, now recognized as GPER, which is structurally different from the classical estrogen receptor (ER) [181,182]. Later, it was discovered that elevated levels of estrogen during the pregnancy phase act upon melanocytes to boost pigment production and melanocyte differentiation [183]. These estrogenic effects are mediated by GPER [183]. In addition, scientists made a hypothesis that studying the relevant hormones, receptors, and downstream signaling proteins activated in melanocytes by pregnancy-related sex hormones could lead to a breakthrough [184]. Accordingly, they discovered that GPER fosters melanoma differentiation, inhibits tumor cell proliferation, and promotes the susceptibility of cancer cells to immune-mediated elimination [184]. Moreover, Natale et al. cloned the oncogene BRAF in human melanoma tissue and grafted the product into mice. Later, the mice were divided into breeding and non-breeding groups. Expectedly, melanoma was not observed in the breeding group as per the hypothesis [184]. In this experiment, they portrayed how depletion of c-myc protein stops the progression of melanoma. However, there is currently no FDA-approved drug acting as a c-myc inhibitor on the global market.

High levels of Myc protein inhibit the genesis of HLA and also increase the production of PDL1, which impedes immune recognition of cancer cells; therefore, T-cells cannot recognize the melanoma cancer cells [184,185]. Scientists also came up with GPER agonists for example, G-1, which showed anti-tumor activity. In their quest for a therapeutic model, they tried GPER agonists and differentiation-based immunotherapy. This opened a new horizon to invent a specific GPER agonist possibly to be used with combination therapy to treat melanoma. Furthermore, they found that the mice with tumors in which the GPER was activated responded better to immunotherapy than those without the GPER treatment. It may be deduced from the experiment that estrogens enhance the immune response of female patients against melanoma [184]. However, further research should be conducted to discover specific GPER-activated drugs with immunotherapy combinations that can serve as better treatment options for melanoma patients. Additionally, scientists also studied the role of GPER in melanogenesis in the human melanoma cell line (A375) and the mouse melanoma cell line (B16) [186]. The authors reported that GPER increases melanogenesis in these cell lines by upregulating microphthalmia-related transcription factor-tyrosinase via activating the cAMP-protein kinase A (PKA) signaling. These findings, therefore, imply that GPER is a potential drug target for chloasma or melanoma [186]. In another study involving mice model [187], it was found that treatment with the GPER agonist G1 of mouse melanoma K1735- M2 cells decreased cell biomass and the number of viable cells without an increase in cell death. Altogether, these studies demonstrate that GPER as a potential target in melanoma therapy.

IHC staining techniques were used to measure the protein expression of GPER in melanoma tumors [188]. To validate the GPER effect on melanoma, a comparative study [189] was carried out with the other classical estrogen receptors (ER-α, β) in common nevi, dysplastic nevi, and melanomas. In this study, GPER expression was substantial in nuclei and cytoplasm of dysplastic nevi and margins compared to melanoma [189]. Most melanomas expressed ERβ and GPER, but the expression was different in margins and sebaceous glands in melanoma and non-melanoma lesions [189]. This points to these receptors as prospective biomarkers. Considering the wealth of evidence that directs the impact of GPER on melanoma cell lines and tumors, we postulate that GPER could be a potential therapeutic target for melanoma patients.

### 3.9. GPER in Cervical Cancer

To date, cervical cancer (CC) has remained by far, a major malignancy affecting women worldwide with human papillomavirus (HPV) considered the major etiologic agent [190]. This is due to the higher prevalence of the virus in most of the women diagnosed with CC. The virus encodes oncogenic proteins (E6 and E7) that are involved in signaling and participates in various cascades of events that ultimately lead to the development of cancer [191]. Infection alone with the HPV is not sufficient to establish CC in the patients, as the interplay of other factors is required to fully dictate the final malignant transformation [192].

Epigenetic modifications and signaling have been considered relevant phenomena in cancer progression [193]. For instance, the E6 and E7 oncoproteins from HPV are associated with the activation and induction of signaling pathways (notably Akt, Wnt/B catenin, and Smad proteins among others) leading to the activation or repression of certain genes implicated in CC [192]. The hormone (estrogen), the function of which is mediated through the estrogen receptors (ERα and ERβ), and the G protein-coupled estrogen receptors (GPER/GPR30) have been associated with the development of CC [129]. Although high expressions of the ER receptors were associated with disease advancement, the activation of the GPER/GPR30, particularly in the tissue, is involved in cancer biology [129]. A study conducted employing qPCR, Western blots, and immunofluorescence techniques showed that, in HeLa and SiHa cells, E6 and E7 oncogenes increased the expressions of ERα, GPER/GPR30, and prolactin receptors (PRLR) and that the oncogenes modified the location of the receptors to the cytoplasm (in the case of PRLR) and nucleus (for the ERs and GPER/GPR30) [192] (Table 3). Hofsjo et al. compared the expression pattern of the sex steroid hormone receptors in CC survivors and non-infected women (control) [194]. Concerning GPER/GPR30, a non-significant difference in the expression of the receptor in both the epithelium and stroma of CC survivors and the control women was observed following immunohistochemistry scoring [194].

Similarly, GPER expression was detected in most tissues collected from CC patients, as well as in different subcellular tissue staining patterns [195]. In the study, there was a positive correlation between cytoplasmic GPER staining and CC tumor suppressor p16 and p53 (but not mutant), although, on the contrary, there was no association between the GPER and E6 and E7 oncogenes [195]. Hence, the study revealed that, in early-stage CC, GPER positivity could be an excellent predictor of overall and progression-free survival [195]. Moreover, there was a negative correlation between Lysine Specific Demethylase 1 (LSD1) and GPER/GPR30 was observed in 250 cervical cancers [196]. LSD1 demethylates lysine 4 of histone H3 serving as a transcriptional co-repressor. Hence the findings reported a disadvantaged 10 years of survival for CC patients with strong LSD1 expressions [196]. Further, mono-2- Ethylhexyl phthalate (MEHP) was found to increases the proliferation of HeLa and SiHa cells and increased the phosphorylation and nuclear localization of Akt in the CC cells, an effect that was reversed in both cells following knockdown of GPER/GPR30 in both cells [197]. Overall, the studies revealed that GPER participated in the compound-induced activation of Akt. Hence, the compound could subsequently trigger the CC progression due to the activation of the GPER/Akt system [197]. These findings implicated GPER in CC epigenetics.

**Table 3 molecules-27-08943-t003:** Summary of Findings on GPER system in Cervical and Thyroid Cancers.

Study-Driven Aim	Methodology	Findings	Conclusion	Reference
Evaluate the mutual regulations of estradiol, and prolactin with HBV E6 and E7 oncogenes as well as the expression and localization of the ERα, GPER/GPR30, and PRLR receptors.	qPCR, Western blot, and immunofluorescence	(a) The hormones induced E6 and E7 expressions. (b) E6 and E7 increased the expressions of the receptors (c) Localization: PRLR (cytoplasm) while ERα, ERβ, and GPER (nucleus).	Mutual regulation exists between the hormones and the oncogenes which leads to an increase in the hormonal receptors and hence could influence CC carcinogenesis.	[192]
Investigate the expressionand distribution of ER, GPER, androgen receptor (AR), progesterone receptor (PR)A, PRB and connective tissue growth factor (CTGF) in the vaginal wall in CC women treated with radiotherapy.	qPCR and immunohistochemistry	(a) lower expression of ERα in CC survivors compared to the normal women (b) ERα, and AR protein expression in the vaginalMucosa significantly reduced, while GPER expression was not affected in CC survivors (compared to normal women) after radiation.	The expression of ERα, and AR but not GPER reduced following radiation in the CC survivors.	[194]
Determine GPER immunopositivity in a cohort of CC patients and investigate its association with other pathological parameters of CC.	Immunohistochemistry	(a) GPER was detected in the tissues of the majority of the CC patients. (b) Cytoplasmic GPER staining was positively correlated to p16 and p53 (non-mutant) and no association was observed between the receptor and HPV E6 and E7.	The study revealed that GPER correlated with HPV-induced tumor suppressor proteins p16 and p53.	[195]
Study the role of LSD1 in cervical cancer patients as well as determine its correlation with GPER/GPR30 receptor	Immunochemistry	(a) A disadvantaged 10-year survival was observed in patients that showed strong LSD1 expressions. (b) A negative correlation between LSD1 and GPER/GPR30 was observed.	Epigenetic changes that could be mediated through the LSD1 and the GPER/GPR30 could serve as an important marker in understanding the expression pattern of the LSD1.	[196]
To study the effects and related mechanisms of MEHP –induced proliferation of CC	Western blotting, Knockdown techniques, and immunofluorescence	(a) MEHP induced the proliferation of the HeLa and SiHa cells as well as increased the phosphorylation and nuclear localization of Akt in the CC cells (b) GPER knockdown reversed the EHP-induced cell proliferation o	Activation of the GPER/Akt system via the action of MEHP could influence the progression of CC.	[197]
To investigate the possible role of GPER in WRO and FRO TC cell lines	Western blotting, MTT, Transwell, ELISA, BrdU incorporation assays	(a) GPER was expressed in the TC cell lines. (b) GPER stimulation was reported to be the initial step for the activation of ERK and AKT pathways, nuclear translocation of NF-kB, and subsequent activation of downstream genes.	GPER signaling pathway (in addition to other related pathways) plays an important role in the progression of TC.	[198]
To access the gene and protein expressions of GPER in PTC and non-malignant thyroid tissues.	qPCR, In silico, and immunohistochemistry	(a) low expression of GPER in PTC which could be associated with BRAF mutation	GPER has a role in PTC development and progression.	[199]

### 3.10. GPER in Thyroid Cancer

Thyroid cancer (TC) is another class of estrogen-related cancer that frequently occurs in women, and, similar to CC, estrogen binds the GPER with high affinity (in addition to ER-α), thereby mediating cascades of reactions leading to the activation of signaling pathways and related genes in TC [198]. For instance, in thyroid cancer cell lines, there is high expression of GPER, similar to MCF-7 breast cancer cell lines. This in turn leads to activation of ERK and AKT pathways, hence, causing increased nuclear translocation of NF-kB and subsequently leading to the activation of downstream genes, including cyclin A, cyclin D1, and interleukin 8 (IL-8) [198]. Bertoni et al. found low expression of GPER in papillary thyroid carcinoma which could be associated with BRAF mutation (PTC) [199]. However, the study further revealed a positive correlation of the GPER mRNA expression level with thyroid differentiation genes [199]. Although a recent review suggested that disease progression could be mediated by the GPER, there were inconsistent findings on the role of the protein in the TC [200]. Overall, the limited studies on the role of GPER in TC suggest the need for a thorough investigation on the role of GPER in the progression of TC. Table 3 summarizes the findings on the role of GPER in CC and TC.

### 3.11. GPER in Ovarian Carcinoma

According to the American Cancer Society, about 12,810 women in the United States will die from ovarian cancer in 2022. Ovarian tumors are classified into three main subtypes: ovarian epithelial carcinomas, germ cell tumors, and stromal cell tumors (sex cord) [201]. These malignancies are differentiated by cell/site of origin, histological appearance, risk factors, and clinical characteristics [202]. Unfortunately, most patients (4 out of 5) with ovarian cancer are diagnosed with advanced disease, and epithelial carcinoma accounts for 85–90% of the cases [4]. Estrogen, the primary sex hormone in females [203], and is involved in breast, endometrial, and ovarian physiology. Mainly, estrogens are synthesized in the ovaries and adrenal glands [204] and regulate the growth and differentiation in the normal ovaries. Moreover, estrogens are involved in the progression of ovarian cancer. There are three physiological subtypes of estrogens: estrone (E1), estradiol (E2), and estriol (E3), with E2 being the major and most active subtype [205]. These three forms of estrogen bind to the GPER [206], which is involved in rapid estrogen signaling. GPER expression varies with age, gender, and tissue. Estrogens binding to GPER induce ovarian cancer cell proliferation through activation of multiple downstream signals such as ERK (extracellular-signal-regulated kinase), PI3K (phosphoinositide 3-kinase), and EGFR (epidermal growth factor receptor) [207]. Additionally, GPER overexpression enhances Akt phosphorylation via EGFR. Both GPER and EGFR overexpression is associated with poor outcome for ovarian cancer [208]. The upstream regulation of GPER is induced explicitly by LH/FSH along with FSHR-GPER dimerization which stimulates FSH-Stimulated Ovarian Follicle viability via Gβγ dimers [209]. Furthermore, GPER regulates the migration and invasion of ovarian cancer cells (SKOV3 and OVCAR5), and targeting of GPER could provide a new therapeutic strategy for ovarian cancer [210,211].

On the other hand, Tanja Ignatov et al., and Wang, Cheng, et al. proved that GPER acts as a tumor suppressor in ovarian cancer, and the expression of GPER was lower in ovarian cancer tissue compared to benign as well as early-stage cancers [212,213]. This finding aligns with the observations made by other researchers that support GPER as a tumor suppressor in ovarian carcinoma [214]. Fraungruber, Patricia, et al. addressed the connections between estrogen and Wnt signaling in ovarian cancer. They showed that the combined expression of GPER and the Wnt pathway modulator Dickkopf 2 (Dkk2) signaling pathway is associated with improved overall survival (OS) [215]. Moreover, Zhu, Cai-Xia, et al. group [216] showed that GPER expression varied in different histological subtypes of ovarian cancer. The same group explained that cytoplasmic localization of GPER is not associated with the outcome, but nuclear GPER predicts 5-year progression-free survival and poor survival for ovarian cancer patients. In contrast, Kolkova, Zuzana, et al. consider that, for patients with ovarian cancer, neither GPER mRNA nor protein predicts survival, and they do not correlate with histological or clinical parameters in patients with ovarian cancer [217]. Therefore, the role of GPER in ovarian carcinoma is still controversial and not clear.

The impact of GPER on cancer cell growth or suppression depends on the cancer type and tissue target. The previous findings demonstrate the complex roles of GPER in ovarian carcinoma that need additional research. In addition, since numerous areas lack knowledge about GPER and its function, further investigation is required. For example, it was reported in one study covering the epigenetic regulation of GPER in ovarian tumors that GPER activation epigenetically regulates the trimethylation of histone H3 lysine 4 (H3K4me3) and ERK1/2, which leads to cell proliferation and migration inhibition in ovarian cancer cells [218].

### 3.12. GPER in Prostate Cancer

Prostate cancer (PCa) is the most common malignancy affecting men. Although the incidence and mortality rates vary worldwide, PCa remains the second-leading cause of cancer death in the US [219]. According to the American Cancer Society, approximately 268,490 new cases of PCa will be observed and 34,500 people will die, in 2022. The most common histological subtype is prostate adenocarcinoma, which can rapidly progress into hormone-refractory prostate cancer (HRPC). The biological heterogeneity of prostate cancer reveals the disease complexity in clinical and research settings [220]. However, several studies imply that alterations of different pathways involving growth factor receptors are involved in prostate cancer aggressiveness [221]. Growth factor receptors (mainly EGFR) and G protein-coupled receptors (GPCRs) mediate a complex signaling network that activates relevant biological effects in cancer cells [222].

Moreover, GPCRs, mainly GPER, regulate multiple biological responses [223], including cancer cell proliferation and migration. Both androgens and estrogens mediate prostate cancer development and differentiation, and their combined effect induces disease aggressiveness [224]. Of note, estrogen receptors (ERα and ERβ) are found sequentially in stromal cells and epithelial lumen cells of the prostate gland [225], and activation of these receptors leads to premalignant and metastatic lesions [226]. Wide-ranging findings on the roles of ERα and ERβ in prostate cancer suggested that ERβ was predominantly protective, and ERα was tumor-promoting [227]. Additionally, GPER is an estrogen mediator detected in male reproductive cells (such as testicular cells and spermatozoa). Still, its expression in prostatic tissue has not been reported yet. The Rago, V., et al. group reported that GPER has strong immunoreactivity in the cytoplasm of basal epithelial cells as well as in Gleason pattern 2 or Gleason pattern3, but no immunostaining was evident in luminal secretory epithelial cells. Even though there was a historical use of estrogens in the pathogenesis of PCa, their biological effect is not well known, nor is their role in carcinogenesis [228]. Ramírez-de-Arellano, Adrián, et al. reported the mechanism of GPER associated with ER and GPER in PCa [228]. Nevertheless, more studies need to be accomplished to explore in depth the role of GPER in prostate cancer prognosis as well as the mechanisms used to carry their therapeutic effects of neoplastic in prostate transformation and the possibility of targeting GPER in advanced diseases in addition to epigenetic regulation of GPER in PCa.

## 4. GPER System in Cardiovascular and Other Pathological Conditions

From its discovery to the present, GPER has been shown to have multifaceted functions in both physiological and pathological conditions. The system expressed in nearly all body tissues, and has been implicated in regulating the differential expressions of various genes and in cAMP generation [229,230,231]. In addition to its roles in the diagnosis and management of diabetes and cancer, this receptor stands as a prognostic indicator and therapeutic target for a number of other human diseases including, but not limited to, reproductive, nervous, cardiovascular, and immune-related diseases [206,232].

### 4.1. Cardiovascular Diseases

The incidence of cardiovascular diseases in premenopausal women is lower compared to their male counterparts, but the latter tend to have a lower incidence than postmenopausal women [233], suggesting the potential inclusion of cardioprotection in the list of estrogen activities. In support of this observation, a study of a rat model found reduced infarct size and improved heart function when GPER was activated by its G-1 agonist [234]. Although the production of 17β-estradiol (an estrogenic ligand of GPER) goes down with menopause, the molecular mechanism behind this GPER function is yet to be fully understood. GPER mutant mice were found to have low risks of cardiovascular diseases due to low levels of superoxide production [235], suggesting a link between GPER function and NADPH oxidase activity.

Hypertension, a prominent risk factor for cardiovascular diseases, also displays sex-related differences as its prevalence is often higher for men, but it becomes more frequent in women above 60 years of age [236,237]. The estrogen 17β-estradiol has a protective role against the development of hypertension (especially angiotensin II-induced hypertension) through modulation of vasoconstriction [238,239]. The activity of this hormone is regulated by GPER, the activation of which lowers blood pressure by promoting vasodilation [240]. Similarly, there is an increase in the expression of GPER in left ventricles of spontaneously hypertensive rats (SHR), compared to the normotensive control group [241]. The authors observed that the elevated expression of GPER elicits a protective effect against hypertension by reducing left ventricular pressure in the hypertensive model. In another study using the SHR heart model, GPER was shown to confer Notch 1-dependent cardioprotection in both hypertensive and normotensive conditions [242]. GPER also modulates coronary tone as inhibition of its function leads to an increase in the coronary perfusion pressure of both male and female rats [243]. As such, GPER can serve as a potential target for the management of cardiac dysfunction.

### 4.2. Reproductive Disorders

Emdometriosis is a female reproductive disorder characterized by the presence of endometrium and stroma in tissues outside the uterus, usually in the pelvis, resulting in lower abdominal pain and sometimes infertility. This gynecologic disorder is estrogen-dependent and, as such, is linked to GPER functions [244]. A previous study revealed an overexpression of GPER in patients with ectopic endometriosis, suggesting its involvement in the regulation and severity of the disease [245]. Consistently, the study discovered a positive correlation between GPER and gankyrin expressions at various stages of endometriosis, suggesting their therapeutic potentials for endometriosis.

Leiomyomas (or uterine fibroids) arises from proliferation of smooth muscle cells of the uterus leading to tumors of the myometrium at childbearing age. The tumors are estrogen-sensitive, and GPER expressions have been recorded for both myometrial and leiomyoma cells, where the receptor was shown to mediate the proliferation through activation of MAPK pathway [246]. The risk of uterine leiomyoma is also attributed to certain SNPs (single nucleotide polymorphisms) that occur in the GPER gene (there are more than a thousand of SNPs in the gene) [247].

### 4.3. Nervous Disorders

The predominant estrogen, 17β-estradiol, exerts diverse effects on both central and peripheral nervous systems, including neuroprotection and regulation of homeostasis, pain stimulus, and synaptic plasticity. Most of these functions are mediated by GPER, though a few may be ERα- and ERβ-dependent [206]. GPER function is linked to amelioration of Parkinson’s disease and global ischemia [248,249]. Parkinson’s disease is a neurodegenerative disorder characterized by a drop in dopamine levels, resulting in an impairment of motor function, which causes uncontrollable movements. In an animal model of Parkinson’s disease, selective activation of GPER reduced lipopolysaccharide-induced microglial reactivity, protected nigral dopaminergic neurons, and prevented impairment of motor neurons [250,251]. Guan and coauthors investigated the relationship between GPER and dopaminergic neuron degeneration induced by the neurotoxin 1-methyl-4-phenyl-1,2,3,6-tetrahydropyridine (MPTP) [252]. Their study found that GPER exerts an anti-neuroinflammatory effect and prevents dopaminergic neurodegeneration in an MPTP animal model of Parkinson’s disease. Together, these results demonstrate the therapeutic potential of GPER in the management of neurodegenerative diseases, particularly in annulling neuroinflammation.

### 4.4. Immune-Related Diseases

Nearly all immune cells express GPER at both early and mature stages of their development. Several independent works have unveiled the roles played by GPER in the development, maturation, and functions of immune cells, notwithstanding the existence of some conflicting findings [253,254]. However, questions remain unanswered regarding the specific roles and functional mechanisms of GPER in immunity. The roles of GPER in each immune cell type (T-lymphocytes, B-lymphocytes, monocytes/macrophages, eosinophils, and neutrophils) have been comprehensively tabulated by Notas et al. [255].

Most of the functions of GPER in various pathological conditions (including malignancy, DM, and hypertension) are the results of its effects in immunomodulation. The neuroprotective effect of the receptor, for instance, is linked to the anti-inflammatory role it exerts on macrophages and lymphocytes [249,252]. In ischemic brain injury, the transmembrane receptor prevents neural injury by blocking TRL4-based inflammation (TLR4 is a member of the pattern recognition receptor family of proteins) [256]. Similarly, G-1-mediated activation of GPER reduces the amount of pro-inflammatory cytokines, thereby ameliorating the severity of multiple sclerosis and encephalomyelitis [257].

Triplett et al. investigated the role of GPER in innate immunity against *Staphylococcus aureus* infection [258]. The team found that activation of GPER using the synthetic and GPER-specific agonist G-1 limits pro-inflammatory cytokine production, which consequently decreases the severity of *S. aureus* skin and soft tissue invasion. Thus, GPER has functions beyond that of an ER. The protein is a potential therapeutic target for many human diseases, and there are apparently undiscovered roles of the protein in relation to other pathological conditions.

## 5. Comparative Perspectives of the Dysregulated GPER System as a Therapeutic Target in DM and Malignancy

Current literature has limited data on the expression patterns of GPER in diabetic models but GPER signaling in high glucose and diabetic conditions results in the altered expression of key transport and signaling proteins. Notably, GLUT-2 is one such protein that shows substantial evidence of overexpression following GPER stimulation [39]. Thus, the protective role of GPER ligands against DM could depend on GLUT-2 expression and function, and its upregulation could be a promising therapeutic strategy. Of note, GPER activation in various tissues seems to be a therapeutic option against high-glucose-induced vascular complications of DM [42]. For instance, the altered expressions of key vascular protective proteins observed in high glucose and diabetic conditions are reversed following GPER activation. TGF-β [43] and eNOS [42] are of particular importance among the GPER-targeted proteins and could serve as therapeutic targets.

The expression patterns of GPER in various cancers are complex and debatable, with some cancers exhibiting GPER upregulation and others having downregulated or even inconclusive GPER expression patterns. For instance, GPER is overexpressed in seminomas [58,59], melanomas [189], some ovarian cancers [208], some lung cancers (NSCLC) [124], and in insulin-resistant endometrial cancer models [168] and in most of the breast cancer (particularly TNBC) [11,66,70,75,101] models reported in this paper. Interestingly, this upregulation is predominantly associated with oncogenic properties. We may, therefore, postulate that GPER itself is a potential therapeutic target in these cancers, and its pharmacological inhibition would be a promising therapeutic intervention. On the other hand, GPER is significantly suppressed in gastric cancers [9] and in some ovarian [75,213], and endometrial cancers [167]. This downregulation is mostly associated with poor prognosis, and hence increased expression or pharmacological activation of GPER could be promising in the treatment of these cancers. Table 4 compares the molecular mechanisms of dysregulated GPER expression in DM and malignancy.

Many signaling pathways downstream of GPER have been altered due to changes in the expression, activation, or inhibition of the receptor in diabetic conditions as well as in cancers (Table 4). However, the most commonly altered pathways in both conditions are the EGFR-ERK/MAPK [77,99,104,107] and the PI3K-Akt/PKB pathways [26,170,173]. The two GPER targets (EGFR and PI3K) are activators or initiators of downstream signaling pathways and thus, could serve as primary foci in the GPER-mediated therapy for most of the cancers we studied and for some diabetic conditions. Paradoxically, although EGFR could be beneficial in T1D, its role in the cancers we studied is mostly oncogenic. On the other hand, PI3K activation seems to be detrimental in both the diabetic and cancer models based on our review, and thus it could be targeted in combined therapy. In addition, the anti-apoptotic protein (BCL2) is a downstream target of GPER in breast cancer [98] and diabetic conditions [44] and thus, could be a therapeutic target for both conditions. Moreover, GPER-mediated upregulation of aromatase has been associated with insulin resistance [26] and could be implicated in breast cancer pathogenesis [106]. Hence, aromatase is among the GPER downstream targets with therapeutic potential for DM and malignancy. We may also suggest here that the dysregulation of signaling proteins downstream of GPER could involve epigenetic reprogramming induced by GPER stimulation. In sum, GPER-mediated attenuation of the EGFR and PI3K signaling pathways could be promising in the treatment of cancer and diabetic conditions as presented in Table 4.

## 6. The Sex-Dependent Function of GPER

According to a recent review, ER**α** or GPER knock-out mice exhibit similar metabolic perturbations such as increased adiposity, decreased insulin sensitivity, impaired glucose/lipid homeostasis, and inflammation. Obesity, diabetes, and cardiovascular disease were found to be less common in premenopausal women than in age-matched men or postmenopausal women. Insulin sensitivity also differs by gender, with premenopausal women being more insulin sensitive than age-matched men or postmenopausal women [23]. Recently, Gene set Enrichment Analysis (GSEA) revealed that mitochondrial genes are enriched in GPER KO females. In contrast, GPER KO males have an increased expression of inflammatory response genes [259]. All these facts point to critical roles of the GPER system within and across genders.

## 7. Pharmacological Interventions on GPER System

GPER, as an indispensable and versatile biomolecule, is a viable therapeutic target in DM and various malignancies, as evident by the pharmacological activities of E2, G1, Tamoxifen, Bisphenol A, Genistein, and others. However, some of the mechanisms of action remain unknown.

### 7.1. Diabetes Mellitus

Streptozotocin-induced diabetic rats treated with E2 at 10 μg/kg for 8 weeks increased GPER with concomitant increase GLUT2 protein and insulin secretion in islet cells via the Akt/mTOR pathway [39]. In broiler chickens, genistein at 25 μM and 150 mg/kg body weight increases glucose catabolism in vitro and in vivo by activating the GPER-mediated cAMP/PKA-AMPK signaling pathway [40]. High glucose-induced vasoconstriction and endothelial damage in OVX db/db mice by RhoA/ROCK/eNOS signaling pathway led to activation of GPER by E2 at 100 nM exvivo and G1 at 200 μg/kg body weight for 8 weeks or 1 nM in vivo or invitro [42]. Similarly, high glucose-induced type IV collagen and fibronectin were found to be reduced in glomerular mesangial cells due to GPER activation with with Icariin at 10–100 μM by inhibiting TGF- production [43].

### 7.2. Malignancies

Tamoxifen at 1 μM continuously exposed breast cancer cells for 7 days upregulates GPER-1 and increases cell proliferation associated with kinin B1 receptor induced signaling [10]. BPA is an estrogenic endocrine disruptor with a low affinity for conventional ERs. However, at very low concentrations (1 nM), it was able to promote seminoma cell proliferation via the GPER system, which was independent of ER pathways [60]. Activating GPER with 10mg/kg body weight of the small molecule agonist G-1 inhibited pancreatic ductal adenocarcinoma proliferation, depleted c-Myc and programmed death ligand 1 (PD-L1), and increased tumor cell immunogenicity in mice [160]. E2 upregulated GPER expression, proliferation, invasion, and migration of breast cancer cells by regulating the miR-124/CD151 pathway at concentrations ranging from 10–100 nM [260]. At the cytoskeletal level, the ER was reported regulates the K19 gene and localizes the estrogen-responsive region, implying that estrogen-induced keratin 19 gene expression may contribute to cytoskeletal and nuclear matrix reorganization and increased metastatic potential in ER-containing breast cancer cells [261]. However, no research has been conducted on GPER and cytoskeletal in breast cancer.

## 8. Future Perspectives

Currently, only a few studies have reported significant alterations in the expression of GPER itself under high glucose or diabetic conditions. Most studies on the antidiabetic role of GPER have focused mainly on the impact of GPER deficiency or its stimulation by a ligand, rather than its expression. Moreover, most of these studies are based on β-cell survival and function. Hence, studies observing changes in the expression patterns of GPER in diabetic conditions, and in various tissues (particularly insulin-sensitive tissues), are warranted to explore how DM affects GPER expression. Furthermore, altered expressions of key genes/proteins involved in insulin secretion and glucose homeostasis following GPER stimulation have been documented under physiological conditions, with little or no evidence for these changes in diabetic conditions, particularly T2D. Further studies on dysregulated GPER in T2D models are, therefore, recommended. However, substantial evidence points towards GPER dysregulation (mostly upregulation) in various cancers, but there is a dearth of data on the molecular mechanisms underlying these changes. We speculate that epigenetic alterations could be involved and hence, we recommend further studies on the epigenetic mechanisms of GPER system dysregulation in various cancers.

## 9. Conclusions

GPER as a therapeutic target has great potential. However, further studies are needed that take into consideration the expression levels of GPER under diabetic conditions, especially in insulin-sensitive tissues compared to those of healthy individuals of the same age and gender. The results from such studies could reveal how GPER expression relates to DM. With respect to GPER and cancers, studies that examine the molecular mechanisms underlying variations in GPER expression such as epigenetic modification could enhance our understanding of the role of GPER in different cancers. Overall, an understanding of the GPER system, especially among patients harboring DM and malignancy, could provide a viable and alternative strategy in the management of the diseases.

## Figures and Tables

**Figure 1 molecules-27-08943-f001:**
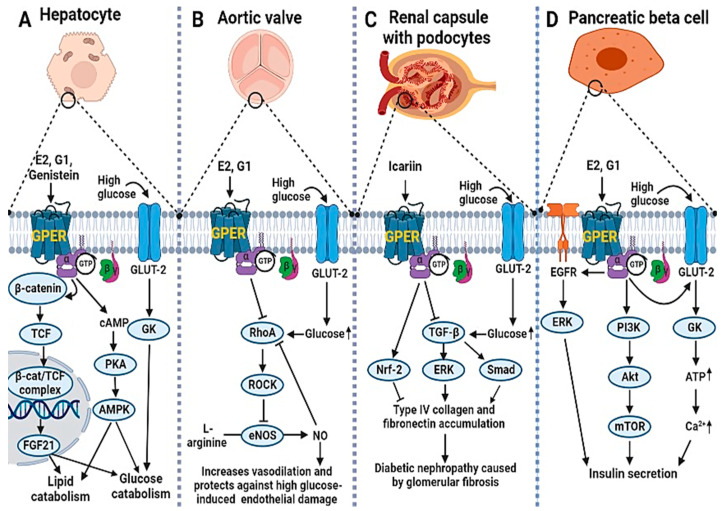
A comprehensive overview of the physiological and pathological changes associated with high glucose-induced alterations in GPER signaling pathways and the protective roles of GPER agonists. (**A**) Genistein enhances glucose and lipid catabolism in high glucose-treated human hepatocytes by activating a GPER-mediated cAMP/PKA/AMPK signaling under physiological conditions. E2 and G1 exhibit a similar effect via GPER-mediated β-catenin/TCF/FGF21 signaling. (**B**) E2 and G1 inhibit high glucose-induced vasoconstriction and endothelial damage in human aortic cell lines and diabetic mice via a GPER-mediated RhoA/ROCK/eNOS signaling. (**C**) Icariin prevents high glucose-induced glomerular fibrosis and diabetic nephropathy in renal tissues of diabetic mice via a GPER-mediated inhibition of TGF-β and activation of Nrf-2 signaling pathways. (**D**) E2 and G1 augment high glucose-stimulated insulin secretion in pancreatic β-cells via a GPER-mediated upregulation of GLUT-2 as well as activation of PI3K/Akt/mTOR and EGFR/ERK signaling pathways under physiological conditions. AMPK = adenosine 5′- monophosphate-activated protein kinase, ATP = adenosine triphosphate, Akt = Ak strain transforming protein/protein kinase B, β-cat = β-catenin, cAMP = cyclic adenosine monophosphate, E2 = 17β-estradiol, EGFR = epidermal growth factor receptor, eNOS = endothelial nitric oxide synthase, ERK = extracellular signal-regulated protein kinase, FGF21 = fibroblast growth factor 21, G1 = a synthetic and selective GPER agonist, GK = glucokinase, GLUT-2 = glucose transporter 2, GPER = G-protein-coupled estrogen receptor, mTOR = mammalian target of rapamycin, NO = nitric oxide, Nrf-2 = nuclear factor erythroid 2-related factor 2, PKA = protein kinase A, PI3K = phosphatidyl inositol 3-kinase, RhoA = ras homolog family member A, ROCK = Rho-associated protein kinase, Smad = suppressor of mother against decapentaplegic. (Source: image was created in BioRender (biorender.com), accessed 25 June 2022).

**Figure 2 molecules-27-08943-f002:**
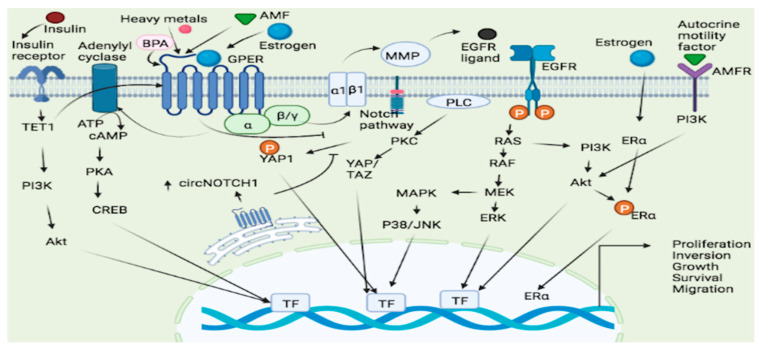
GPER signaling pathways in lung and endometrial cancer. Several ligands such as Estrogen, heavy metals and AMF bind to the GPER leading to the activation and subsequent cellular proliferation, inversion, growth and survival via the GPER/cAMP/PKA/CREB, GPER/RAS/RAF/MEK/ERK and GPER/ERs/ERK/MAPK pathways. Estrogen may also bind to both ER and GPER and generated a signal which leads to increased cell proliferation. GPER-1 interacts with AMF and promotes PI3K signaling resulting in cancer cell growth. TET1 played a key role in upregulating GPER expression and also activating the PI3K/AKT signaling pathway, thus making insulin as a stimulator of EC cell proliferation through TET1-mediated GPER upregulation. ATP = Adenosine Triphosphate, cAMP = cyclic Adenosine monophosphate, YAP = Yes-Associated Protein, PKA = protein Kinase A, PKC = Protein Kinase C, Akt = Protein Kinase B, AMF = Autocrine Motility Factor, AMFR = Autocrine Motility Factor Receptor, ERK = extracellular signal-regulated protein kinase, CREB = cAMP response element binding protein, BPA = Bisphenol A, MAPK = Mitogen-activated protein kinase, PI3K = phosphatidyl inositol 3-kinase, TET1 = *tet methyl cytosine dioxygenase 1, RAF = Rapidly Accelerated Fibrosarcoma, RAS = rat sarcoma, MEK = mitogen-activated protein kinase kinase, ERK =* Extracellular signal-Regulated Kinase, EGFR = *Epidermal growth factor receptor, JNK = c-JUN N-terminal kinase,* ERα = Estrogen receptor α.

**Table 1 molecules-27-08943-t001:** Pattern of GPER Expression Observed in Different Breast Cancer Phenotypes.

S/No	Breast Cancer Phenotype	Experimental Model	Pattern of GPER Expression Observed	Observations	Reference
1	ER-positive breast cancer	MCF-7 cells and ER-positive breast cancer patients	Elevated	Increased tamoxifen resistance and metastasis	[66]
2	ER-positive breast cancer	MCF-7 cells	Elevated expression using the G-1 agonist	Reduced cell proliferation, cell cycle arrest at M-phase and enhanced apoptosis	[70]
3	ER-negative breast cancer	MDA-MB-231 cells	Elevated expression in the presence of bisphenol A (ligand)	Increased invasiveness and metastasis	[71]
4	ER-negative breast cancer	MDA-MB-231 cells	Inhibition via interaction with Na^+^/H^+^ exchanger regulatory factor 1	Inhibited ERK1/2 and suppressed proliferation of the TNBC cells	[72]
5	ER-negative breast cancer	HCC70 and HCC1806 cells	Downregulated via treatment with gefitinib	Suppressed proliferation of the TNBC cells	[73]
6	ER-negative breast cancer	MDA-MB-231 xenograft tumors	Elevated via treatment with G1 agonist	Inhibited angiogenesis and metastasis	[11]
7	ER negative breast cancer	MDA-MB-231 MDA-MB-468	Elevated via treatment with G1 agonist	Inhibited cell growth, induction of cell cycle arrest, increased apoptosis	[70]
8	ER negative breast cancer	TNBC patients in China	Increased expression	Reduced lymph node metastasis	[75]
9	Primary invasive breast cancer patients	Cohort of breast cancer patients in the UK	Low expression	Aggressive disease progression and adverse survival	[76]

**Table 2 molecules-27-08943-t002:** GPER genes implicated in the progression of breast cancer malignancies.

S/N	Gene	Gene Product	Mechanism/Function	References
1.	β1-integrin	β1-integrin	Tumor progression inflammation, proliferation, adhesion, invasion, and tumor survival	[107]
2.	c-FOS	c-FOS	Cell proliferation and migration	[67,107]
3.	EGR-1	EGR-1	Cell proliferation and migration	[67,107]
4.	CTGF	CTGF	Cell proliferation and migration	[94,108]
5.	CYP19A1	Aromatase	Proliferation and cell-cycle progression	[104,109]
6.	FAK	FAK	Differentiation, proliferation, apoptosis, migration, invasion, survival, and angiogenesis	[101]
7.	ERK2	ERK1/2	Cell migration, proliferation, angiogenesis, and invasion	[67,101]
8.	Src	Src	Cell cycle progression, growth, survival and migration	[101]
9.	HIF1α	HIF1α	Endothelial tube formation	[110]
10.	VEGF	VEGF	Endothelial tube formation	[110]
11.	Endothelial marker (CD34)	Endothelial marker (CD34)	Endothelial tube formation	[110]
12.	HOTAIR	HOTAIR	Unknown	[111]
13.	FASN	FASN	Energy production, cell proliferation, aggressiveness, and metastasis	[112]
14.	miR-338-3p	miR-338-3p	suppress the growth and invasion of SkBr3 cancer cells and CAFs	[113]
15.	miR144	miR144	increase cell cycle progression	[113]
16.	onco-suppressor Runx1	onco-suppressor Runx1	increase cell cycle progression	[113]

Footnote: RUNX1= Runt-related transcription factor 1, HOTAIR = HOX antisense intergenic RNA, FASN = Fatty acid synthase, FAK = Focal adhesion kinase, HIF1α = Hypoxia-inducible factor 1 α, SRC = SRC Proto-Oncogene, Vascular endothelial growth factor, VEGF, FASN = Fatty acid synthase, CTGF, Connective Tissue Growth Factor, *ERK =* Extracellular signal-Regulated Kinase, EGR-1 = Early growth response protein 1, CYP19A1 = Cytochrome P450 Family 19 Subfamily A Member 1, c-FOS *= Cellular oncogene.*

**Table 4 molecules-27-08943-t004:** Dysregulated GPER expression in DM and malignancy.

Nature of GPER Alteration in Diabetic Conditions	Consequences of GPER Alteration in Diabetic Conditions	Nature of GPERAlteration in Malignancies	Consequences of Alteration in Malignancy
(a) Differential GPER gene expression in the livers of T2D patients [22] (b) Upregulated GPER gene and protein expression in mice model of obesity and T2D [34] (c) Upregulated GPER protein expression in a T2D rat model following E2 treatment [39]	(a) Could be implicated in hepatic insulin resistance [22] (b) Loss of aldosterone-mediated protection against diabetes-induced endothelial damage [34] (c) Increases GLUT-2 protein content and insulin secretion in islet β-cells [39]	(a) GPER is overexpressed at mRNA and protein levels in seminomas [58,59] (aii) SNPs at GPER gene promoter in seminomas [59] (b) GPER is overexpressed at protein levels in ovarian cancer [208] (c) Downregulated GPER protein expression in ovarian cancer compared to benign and low-malignant ovarian tumors Tanja Ignatov et al. [75] (di) GPER expression is upregulated in TNBC [67,71,75] (dii) GPER gene promoter is hypermethylated in MCF-7 (ER-positive breast cancer cells) [70] (e) GPER gene expression is upregulated in NSCLC [124] (fi) GPER mRNA and protein levels are downregulated in gastric cancer [9] (fii) SNPs along the GPER gene sequence of gastric cancer patients [139] (g) Insulin induces the upregulation of GPER mRNA and protein levels in insulin-resistant endometrial cancer cells by upregulating TET-1 [168] (h) GPER protein is differentially expressed in different melanocytic lesions [189] (I) Upregulated LSD1 protein expression correlates negatively with GPER protein expression in cervical cancer patients [196]	(a) Stimulation of seminoma cell proliferation by activating ERK1/2 and protein kinase A [58,59] ERβ downregulation causing a switch in estrogen responsiveness [63] (aii) Correlate positively with GPER upregulation in seminomas [59] (b) Causes poor progression-free survival in ovarian cancer patients [208] (c) GPER could be a tumor suppressor in ovarian cancer Tanja Ignatov et al. [75] (di) Contributes to TNBC invasiveness, metastasis, and recurrence by activating FAK and STAT3, which alter the expression of target genes such as *ERK2*67,72; Reduce lymph node metastasis in TNBC patients [75] (dii) Reduced GPER gene expression and its tumor suppressive role in the studied cell lines [70] (e) Promotes NSCLC cell proliferation [124](fi) Predicts poor prognosis in gastric cancer [9] (fii) Correlate with gastric cancer propensity, advancement, and severity, particularly among the Asian population [139] (g) UpregulatedGPER predicts poor prognosis and stimulates endometrial cancer cells proliferation [168] (h) GPER could be a prognostic biomarker in melanoma [189] (I) Disadvantaged 10-year overall survival in the studied patients [196]

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
