# Peer review of "Comparative G-Protein-Coupled Estrogen Receptor (GPER) Systems in Diabetic and Cancer Conditions: A Review"

_molecules, 2022, doi:10.3390/molecules27248943_

Round 1

Reviewer 1 Report

Dear Author,

This is an interesting article.

Here are my observations/questions/comments:

1.    Abstract – I suggest you mention regarding at least the first statement “diabetes mellitus” since there are different types of diabetes

2.    The same observation as no.1 concerning the main text

3.    Abstract and text body – I suggest to alternatively use the term “malignancy” instead of “cancer”, “cancers”

4.    Lines 40-41 – There is also secondary diabetes mellitus as seen in Cushing syndrome, acromegaly, etc.

5.    Line 60 - The number is not clear

6.    Line 72 - You started with a result of one study before introducing the methods of your research. I suggest a general introduction then provide the particular data after methods of research are provided

7.    You need a section of Methods in order to provide the selection of cited papers, their level of statistical evidence, etc. Also, you should mention as study design that this article is a narrative review of literature and main key words of research are necessary to be specified.

8.    Figure 1 – Is this figure original or a reproduction?  The same for Figure 2.

9.    GPER system and oestrogens pathways: are there any gender differences (for instance between males and females, including pre/post-menopausal status?)

10. Please decide if you use ”BC” for breast cancer and introduce it from the start  

11. Malignancy and GPER – I suggest you first introduce the general connection then provide subsections concerning different types of malignancies (chapter 3, before subsection 3.1.)

12. Is there any importance of skeleton spreading (metastasis) with regard to oestrogens/GPER status?

Best regards,

Author Response

We thank the reviewers for his/her positive comments regarding our manuscript that: "This is an interesting article."

We also value the reviewer's additional recommendations for improving the manuscript. We have responded to the reviewer's additional comments below:

Reviewer #1

Responses to statements 1-6: Throughout the manuscript, we changed diabetes to diabetes mellitus and cancer to malignancy. According to the reviewer's suggestions, we modified the sentences in lines 40-41 and clearly stated the numbers in line 60.

Response to Statement 7: We included methodology.

Response to Statement 8: Figures 1 and 2 are originals, not reproductions.

Statements 9 response: We included a separate section on GPER's sex-dependent role.

Statements 10-12 are consistently used as breast cancer and removed BC. In Chapter 3, we provided a general overview of cancers. We also mentioned that no research on GPER and cytoskeletal in breast cancer had been conducted under the breast cancer section.

Our manuscript was checked by a native English-Speaking Colleague.

Reviewer 2 Report

This is a rather lengthy, but detailed review describing G protein-coupled estrogen receptor (GPER) expression and signaling pathways involved in diabetes and cancer. The general conclusion is that the role of GPER signaling pathways in diabetes and cancer is still rather poorly understood, but warrants further investigation, which includes exploring the therapeutic potential of targeting GPER biology.

My suggestion to improve the manuscript is to include a separate section that critically describes the pharmacology of GPER. Throughout the text one  refers to studies that use diverse ligands/molecules (including E2, G1, tamoxifen, Bisphenol A, icariin, genistein, herbicides, pesticides...), for which it is, however, not always entirely clear to what extent they specifically act on GPER (if known), or if (some of these) ligands also activate/inhibit the classical estrogen receptors (ERa and ERb). This is important information and it might be helpful to the reader to have this information available in a separate section or paragraph. 

A minor comment is that the authors should carefully check the references. For instance, ref. 212, 239 and 255 are identical.

Author Response

We thank the reviewers for his/her positive comments regarding our manuscript that: "This is a detailed review."

We also value the reviewer's additional recommendations for improving the manuscript. We have responded to the reviewer's additional comments below:

Reviewer # 2

We included a separate section on “Pharmacological Interventions on GPER System” and removed duplicate references.

Our manuscript was checked by a native English-Speaking Colleague.